# High Temperature Alters Phenology, Seed Development and Yield in Three Rice Varieties

**DOI:** 10.3390/plants12030666

**Published:** 2023-02-02

**Authors:** Pranee Sanwong, Jirawat Sanitchon, Anoma Dongsansuk, Darunee Jothityangkoon

**Affiliations:** 1Department of Agronomy, Faculty of Agriculture, Khon Kaen University, Khon Kaen 40002, Thailand; 2Salt Tolerance Rice Research Group, Khon Kaen University, Khon Kaen 40002, Thailand

**Keywords:** high temperature stress, physiology, seed development, embryonic development, seed quality

## Abstract

Rice is an important and main staple food crop. Rice in Thailand grows in both the on- and off-seasons. The problem of growing rice in the off-season is that it is dry and the temperature tends to be high. To evaluate the effects of high temperatures on their phenology, yield and seed quality, three rice varieties were cultivated off-season in 2018 and 2019. Rice plants were grown in cement pots on planting date I (PDI; off-season; mid-January) and planting date II (PDII; late off-season; beginning of February). The results showed that rice plants were exposed to higher temperatures in 2019 (than 2018), as indicated by a higher accumulated growing degree day (AGDD). The high AGDD affected the phenology of the rice by shortening the duration of its development from sowing to physiological maturity (PM) from 106.8 DAS in 2018 to 86.0 DAS in 2019. The high AGDD shortened the development duration of the embryo and endosperm, resulting in reductions in the size and growth rates of the embryo and endosperm, and eventually reduced the yield and the yield components. Moreover, the high AGDD reduced the seed quality, as indicated by a decline in the seedling growth rate (SGR) and an increase in chalkiness. Among the varieties, the high temperature in 2019 caused the smallest phenological shift in Chai Nat 1 (CN1), while the shift was largest in Pathum Thani 1 (PTT1). In addition, CN1 exhibited a significantly higher total seed weight/panicle, 1000-seed weight and percentage of filled seed/pot than SP1 and PPT1. It was suggested that CN1 could be described as heat tolerant, and PTT1 as heat sensitive. It was also suggested that farmers should select appropriate rice varieties to grow in the off-season due to the risk of a high-temperature-induced reduction in the seed yield and quality.

## 1. Introduction

Climate change has many impacts on and adverse consequences for agriculture. The Intergovernmental Panel on Climate Change (IPCC) predicted that the mean global air temperature would increase 1.4–5.8 °C in the 21st century, and this is the main cause of global warming [1]. High temperatures decrease crops, livestock and fishery yields by 10.0–25.0% [2,3]. Rice (*Oryza sativa* L.) is one of the main crops and the staple food for over 50.0% of the world’s population. Thailand is the world’s fourth largest rice producer, with 11 million hectares devoted to growing rice and a production quantity of 31.5 million tons in 2019/20. It is also the world’s second largest rice exporter, contributing more than 22 % of the global rice trade volume [4]. Rice in Thailand is obviously grown in two cropping periods, including the wet and dry seasons. Rice that is cultivated during the off-season in Thailand (during January to April [5]; the driest months of the year in Thailand) is called dry season rice or off-season or summer rice. The cultivation of off-season rice is becoming increasingly popular in Thailand, and the off-season rice area increased by 42.9% in from 2016 to 2020. In 2020, the rice yield in the on-season in Thailand was 26,423,822 tons, and the yield in the off-season was 4,553,778 tons. In northeastern Thailand, in particular, the rice growing area in the off-season increased by 65.4% [6,7]. The cultivation of rice in the off-season has a high potential to increase the rice yield, with rice yields of approximately 631 kg/rai, by 43.4%, which is higher than that of on-season rice (approximately 440 kg/rai) [6]. Additionally, Thai farmers can grow and harvest a second crop each year, which is useful to improve overall incomes and food security. However, the problem of rice growing in the off-season is that the prices are lower prices than in the in-season, but the demands for rice production are higher. Moreover, the growing rice in the off-season has to face dry weather and high temperatures during hot summer condition that cause a negative impact on rice growth and development, and also on grain and seed yield quality.

Rice is best grown in a tropical climate where the temperature is between 25.0 and 32.0 °C and the temperature threshold is 35.0 °C [8]. In the northeast of Thailand, the off-season for rice production is from January to May, and the temperature of the hottest month, April, may exceed 40.0 °C. High temperatures may coincide with the seed development stage at a late planting date, affecting the yield and quality of the rice. They also affect the rice’s phenology, physiology, seed development and yield [9,10]. Rani and Maragatham [11] demonstrated that an increase in temperature of 2.0 or 4.0 °C over the ambient temperature reduced the rice growing period by 6.0 or 12.0 days, respectively. Additionally, high temperatures decreased the rice grain filling period from 32.0 to 26.0 days [12].

The exposure of rice crops to high temperatures for 1.0 or 2.0 h during anthesis increased the percentage of grain sterility [13]. High temperatures reduced the pollen viability, pollen germination on stigma and abnormal anther dehiscence, causing sterility and productivity loss [14,15]. High temperatures induced high percentages of spikelet sterility, lower spikelet fertility and reduced grain-filling [10,13], resulting in fewer filled grains, a lower grain weight per panicle and a decreased harvest index [10]. For the heat-sensitive IR64 rice, a high temperature of ≥33.7 °C for over 1 h at anthesis caused grain sterility [16]. When the mean temperatures increased from 35.0 to 45.0 °C, the percentage of sterile grains increased up to 100% [17]. Moreover, high temperatures also reduced the photosynthesis rate in the middle of the ripening phase by 40.0–60.0% and sped up flag leaf senescence [18]. The effects of a high temperature during the grain-filling phase decreased starch synthesis, increased starch degraded enzymes (amylase), inhibited starch assimilation and increased grain chalkiness [19]. The effects of high temperatures also interfered with the assimilation of carbohydrate and grain proteins [20].

High temperatures have been reported to affect grain quality. An increase in the percentage of chalky grain per unit area was found to correspond with an increase in temperature during the grain filling period [18,21,22,23]. Increased temperature during seed development and maturation can affect seed germination [24]. In particular, high temperatures at physiological maturity can reduce the seed vigor and seed germination [25]. Rice seeds exposed to 39.0 °C for 24.0–72.0 h during early seed development resulted in seeds failing to germinate [26]. During seed development, temperatures between 28.0–34.0 °C not only affected seed germination but also affected seed storability, giving a 55.0% shorter shelf life than normal growth [27]. Most of the studies on the effect of high temperatures on rice production have been conducted in the temperate zone. Therefore, this study aimed to elucidate the influence of high temperatures on the phenology, seed development, seed yield and grain quality of rice grown with yearly variation. Useful information obtained from this study will be used for decision making about choosing rice varieties and can be a guideline for rice production in a dry season in Thailand or neighboring tropical countries, while contributing to the adaptation of farmers to climate change and alleviating the impact of high temperature conditions.

## 2. Results

### 2.1. Climate Data

Climate data for three rice growing periods (sowing-PM) under field conditions were observed during 15.0 to 124.0 days of year (DOY) (15 January to 3 May) for PDI and during 32.0 to 137.0 DOY (1 February to 16 May) for PDII in 2018. In 2019, data were collected between 15.0 and 103.0 DOY (15 January to 12 April) for PDI and between 32.0 to 116.0 DOY (1 February to 25 April) for PDII in 2019 as shown in Figure 1. The average rice growing time course (sowing-PM) of PDI and PDII in 2018 was 109.0 and 105.0 days, respectively, and that of PDI and PDII in 2019 was 88.0 and 84.0 days, respectively (Figure 1).

During the experimental period, the highest T_max_ on PDI and PDII in 2018 was 40.0 °C and the highest T_max_ on PDI and PDII in 2019 was 40.4 °C and 41.9 °C, respectively (Figure 1A,D). The average daily T_aver_ on PDI and PDII in 2018 was 26.2 °C (Figure 1B) and 26.5 °C (Figure 1E), respectively. In 2019, the average daily T_aver_ on PDI and PDII for the same year was 30.1 °C (Figure 1B) and 30.8 °C (Figure 1E), respectively. The lowest T_min_ on PDI and PDII in 2018 was 11.8 °C, and, in 2019, the lowest T_min_ on PDI and PDII was 17.0 ^o^C (Figure 1C) and 20.0 ^o^C (Figure 1F), respectively.

Moreover, the average RH on PDI and PDII in the same year was 49.10% (Figure 1A–C) and 49.20% (Figure 1D–F), respectively. In 2019, the average RH on PDI and PDII in the same year were 27.40% (Figure 1A–C) and 25.40% (Figure 1D–F), respectively. These climate data during the rice growing period showed that the average T_max_ on PDI and PDII in 2019 increased by 4.1 °C (12.96%) and 4.4 °C (13.79%), respectively, compared to 2018. In addition, the average T_min_ on PDI and PDII in 2019 increased by 3.6 °C (17.59%) and 4.1 °C (19.60%), respectively, compared to 2018. The overall climate data for the rice growing period showed that the temperature in 2019 was higher than in 2018, and the RH was lower in 2019 than in 2018.

### 2.2. High Temperature Influenced Rice Phenology Shift

The observation of temperature indicated by the accumulated growing degree days (AGDD) under field condition was found to be an average of 1912.1 °C (in 2018) and 1883.8 °C (in 2019) during the three rice growing periods (sowing-PM) as shown in Figure 2A. Both the AGDD of 2018 and that of 2019 affected the period of rice growing from sowing to PM by 106.8 and 86.2 days, respectively (Figure 2A). In 2018, the average AGDDs in PDI and PDII were 1927.8 °C (Figure 2B) and 1896.4 °C (Figure 2C), respectively, which influenced the rice growing time course by 109.0 (Figure 2B) and 105.0 (Figure 2C) days, respectively. The average AGDDs in PDI and PDII in 2019 were 1865.0 °C (Figure 2B) and 1902.5 °C (Figure 2C), respectively, impacting the rice growing period by 88.0 (Figure 2B) and 84.0 (Figure 2C) days, respectively. Thus, the AGDD in 2019 was higher than that in 2018 due to the higher temperatures in 2019 (Figure 1). The higher AGDD resulted in the shortening of all rice growing periods during the vegetative to PM phase. Thus, a higher accumulated temperature in the rice growing time course decreased the rice phenology shift. In the same way, the AGDD in PDII in both years influenced a phenology reduction (Figure 2C).

The AGDD that influenced the rice growing in each variety (Pathum Thani1 (PPT1), Suphan Buri1 (SP1) and Chai Nat1 (CN1)) was higher in 2019 than in 2018 (Figure 2D–I). The average AGDDs in 2018 and 2019 (approximately 720.83 and 952.42 °C, respectively) impacted the phenology shift at germination-PI for all rice by 43.96 and 46.25 days, respectively (Table 1). This showed that the higher AGDD in 2019 caused an extended period of vegetative development in these rice varieties. During PI-milky, the AGDDs in 2018 and 2019 were found to be approximately 1414.03 and 1640.85 °C, respectively. In contrast to the scenario in the vegetative phase, the higher AGDD in 2019 resulted in a reduction in the duration of development from PI to milky from 38.15 to 29.5 days (Table 1). Therefore, the higher AGDD in 2019 caused an average of a 8.57-day reduction in the PI-milky development period compared to that in 2018 (Table 1). The duration of development from milky to PM was 11.4 days in 2019, while that in 2018 was 14.1 days (Table 1). This was associated with the higher AGDD in 2019 (1901.68 °C) than that in 2018 (1688.58 °C). Therefore, the higher AGDD in 2019 influenced extended phenology at the vegetative stage (germination-PI). In contrast, during the reproductive stage (PI-milky and milky-PM), the higher AGDD was associated with a shortening of the developmental period.

### 2.3. High Temperatures Influenced Embryo and Seed Development

The average temperature indicated by the AGDD in each of the rice vars. PPT1, SP1 and CN1 is shown in Figure 3. The average AGDD in 2019 at sowing-PM was approximately 1757.8 °C for PPT1, 1953.3 °C for SP1 and 1940.4 °C for CN1. The AGDD in 2019 for each rice variety influenced the reduction of the growing period at the sowing-PM, and at the sowing-PI in all rice excluding CN1 planting on PDII, which increased by 47.0 days compared to 2018 (Figure 3F). In addition, the AGDD in 2019 reduced all the rice growing periods at PI-anthesis and at anthesis-PM compared to 2018 (Figure 3A-F). In particular, we found that PTT1 showed the highest decrease in the growing period at sowing-PI, PI-anthesis and anthesis-PM, which decreased by 21.0, 21.0 and 27.0 days, respectively. Thus, PTT1 was more sensitive to the higher AGDD than the other varieties.

In all three rice varieties, the number of days to reach anthesis was sharply reduced in 2019 compared to the number in 2018 (Figure 4 and Table 2). The high AGDD in 2019 of approximately 1359.0 °C exercised the highest influence on the embryo and endosperm development in PTT1 by 64.0 days after anthesis (DAA) (Figure 4A,B and Table 2) compared to others.

For the embryo development in all rice varieties, the average AGDD at 1489.9 °C at the anthesis stimulated the globular stage (S1) (Appendix A) at 69.7 days after sowing (DAS) in 2019, which was faster than in 2018 (85.2 DAS with average AGDD of 1473.2 °C) (Table 2). We found the maturation stage (S4) formation at 84.8 DAS with the average AGDD of 1827.3 °C in 2019 also occurred more rapidly than in 2018 (100.2 DAS with average AGDD of 1778.9 °C). This resulted in the growth rate of the embryo formation being markedly higher in 2018 (0.28 mm^2^/day) than 2019 (0.25 mm^2^/day), and the embryo size in 2018 was significantly larger (3.40 mm^2^/seed) compared to that in 2019 (3.08 mm^2^/seed) (Table 3). In 2018 and 2019, the size and growth rate of maturation stage (S4) (Appendix A) in PTT1 exposed to the AGDD were markedly greater than those of the other varieties.

For endosperm development in all the rice varieties, the average AGDD of 1489.9 °C at the anthesis stimulated the initial endosperm development at 69.7 DAS in 2019, which began more rapidly compared to 2018 (85.2 DAS with AGDD of 1473.1 °C) (Table 2). We found the maturation of the endosperm at 86.2 DAS with the average AGDD of 1883.8 °C in 2019, which was more advanced than in 2018 (106.8 DAS with average AGDD of 1912.1 °C) (Figure 4 and Appendix A). This resulted in a growth rate of endosperm formation that was markedly higher in 2018 (0.56 mm^2^/day) than in 2019 (0.48 mm^2^/day), and the endosperm size in 2018 was greater (14.82 mm^2^/seed) compared to 2019 (13.69 mm^2^/seed) (Table 3). Additionally, the endosperm size in 2019 was markedly greater in PTT1 than in the other varieties, and SP1 showed a higher endosperm growth rate than the others (Table 3).

### 2.4. High Temperatures Influenced Yield Components and Yield

We examined the impact of the temperature indicated by the AGDD on the yield components and the yield of the three rice varieties in 2018 and 2019 as shown in Table 4. We found that the planting date in both years did not influence the yield and yield components, excluding the 1000-seed weight, in all rice varieties. In 2019, the AGDD of approximately 1883.8 °C had more of an influence, leading to a significant reduction in the seed weight/panicle (−53%), 1000-seed weight (−9.1%), total seed weight/pot (−41.8%) and filled seed/pot (−30.4%) in all rice varieties compared to 2018 (AGDD = 1912.1 °C) (Table 4). As for CN1 in 2019 (indicated as high temperature), it exhibited higher values for filled seed/pot, total seed weight/panicle and 1000-seed weight than PTT1 and SP1. However, at the same condition, CN1 was the lowest in no. panicle/pot and no. panicle/plant compared to others (Table 4). PTT1 and SP1 in 2019 were not significantly different in no. panicle/pot, no. panicle/plant, filled seed/pot, total seed weight/panicle and 1000-seed weight.

### 2.5. High Temperatures Influenced Grain and Seed Quality

The quality of the rice grains and seeds growing under pot field conditions was determined in 2018 and 2019 as shown in Table 5. We found that the temperature indicated by AGDD in 2019 (1883.8 °C) influenced a reduction in the grain quality, such as the seed width and seed area, and that it induced more chalky grains compared to the AGDD in 2018 (1912.1 °C). In addition, this AGDD in the same year affected the seed quality by significantly reducing the germination percentage (G), germination index (GI), accelerated ageing test (AA) and seedling growth rate (SGR) compared to 2018 (Table 5). The AGDD for both planting dates in the year 2019 induced high chalky grain but reduced the SGR in all rice varieties. The AGDD in PDI during growing period (1896.4 °C) induced a markedly higher G, GI, AA and SGR than the AGDD in PDII (1899.5 °C). Surprisingly, the AGDD in the growing period of CN1 in 2019 influenced the grain quality, such as chalky grain, and seed quality, such as the AA and SGR, compared to PP1 and SP1 (Table 5). This suggests that the higher AGDD affected the grain and seed quality indicated by increasing the grain chalkiness and decreasing the AA and SGR in all rice varieties.

### 2.6. Relative Level of Phenology, Seed Development and Yield Traits in Rice under High Temperatures

Principal component analysis (PCA) was performed to visualize the similarities and differences in the 25 quantitative traits, including the phenology, seed development and yield traits, of the rice due to the high temperatures indicated by the AGDD (Figure 5). The 25 quantitative traits clustered tightly and showed a related shift in the 25 quantitative traits of the rice under high temperatures on PCA plot (Figure 5). A clear separation of the different 25 quantitative traits was observed along with the first principal (PC1 = 40.64%) and the second principal (PC2 = 16.08%). It is noticeable that there were nine traits components that mainly contributed to the highest variation in PC1, which was positively associated with the anthesis day, AGDD of anthesis, endosperm growth rate, total seed weight/panicle, 1000-seed weight, seed length, seed area and chalky grain, excluding accelerating ageing, and five that traits contributed to PC2, which was positively related to the germination-PM, milky-PM, embryo growth rate and no. panicle/pot.

The nine traits with high variability were expected to provide a high level of high temperature tolerance with high genetic variability in rice. Thus, we considered the high temperature tolerant traits of the three rice varieties through a combination of PCA (nine traits in PC1) and distinctive traits (eight traits: embryo size, embryo growth rate, endosperm size, filled seed/pot, no. panicle/pot, no. panicle/plant and SGR) (Figure 5 and Table 6). Consequently, we found that CN1 exhibited the highest high temperature tolerance, indicated by the highest level of anthesis day, AGDD of anthesis, total seed weight/panicle, 1000-seed weight, phenology (sowing-PM), filled seed/pot and SGR, and was then followed SP1 and PTT1. However, we also found that CN1 exhibited the highest in percentage of chalky grain.

## 3. Discussion

The rice plants were grown under different temperatures in different years (in 2018 and 2019) and with different transplanting dates in the off-season (PDI) and late off-season (PDII) in the same field. The elevated temperature during the rice growth was the most important factor that caused either delays or advances in the plants’ phenological development [28,29]. The increasing temperature accelerated changes in the rice phenology, such as panicle initiation, anthesis and maturity. From our results, the AGDD in 2019 was higher than that in 2018, which affected the period of rice growth from sowing to PM by shortening the phenology shift to 86.2 days in 2019 compared to 106.8 days in 2018. Rice at the reproductive stage (PI-milky and milky-PM) was affected by the higher AGDD in 2019, which decreased the phenology shift. This was similar to the effects on the development of anthesis and maturity dates seen in various other agricultural crops, such as wheat [28,30,31], cotton [30] and rice [31].

An increase in temperature of 10 °C shortened the growing period of Basmati rice by approximately 3 days and, at a temperature of 32.0 °C, the Basmati rice seed development period decreased from 32 to 26 days after flowering, and the grain weight was reduced by 6.1% [12]. Similar results were reported by Rani and Maragatham [11]: increasing the temperature by 2 and 4°C above the ambient temperature reduced the crop duration by 6 and 12 days, respectively. However, at the vegetative stage (germination-PI), our results showed that a higher AGDD influenced an extended phenology shift in the rice (Table 1).

The higher AGDD shortened the rice’s life cycle, resulted in a decreased number of days to anthesis (DAA), and also shortened the embryo and endosperm development periods. The higher AGDD reduced the embryo size, embryo growth rate, endosperm size and endosperm growth rate in all rice varieties. This suggests that, under high temperatures, after fertilization, the rice zygote cell and endosperm nucleus divided immediately [32,33,34] by mitosis cell division. That resulted in inhibited tobacco cell division. Thus, in the case of rice, high temperatures may inhibit mitotic cell division during embryo and endosperm development by interrupting mitotic spindle microtubules or phragmoplast microtubules. That would result in a reduction in the number of rice embryos and endosperm cells and, consequently, in reduced embryo and endosperm development. Additionally, the size and growth rate of the rice endosperm decreased due to the high temperatures, resulting in inhibited endosperm development. This affected the sink ability of the rice at the grain filling stage, because there were a limited number of endosperm cells (due to inhibited cell division) to store photoassimilates transporting from the source to the sink cells (endosperm cells). As a result, there were numerous unfilled grains under high temperature stress. There was a report that showed that a medium temperature (34 °C) and a high temperature (42 °C) affected endosperm development at an early stage. A moderate temperature of 34 °C reduced the seed size, and a high temperature of 42 °C induced incomplete rice endosperm development. This indicated that a high temperature decreased the seed growth rate at the PI-maturity stage and inhibited starch accumulation in the grain at the grain filling stage, according to Folsom et al. [35].

In the case of PPT1 under a high temperature, it exhibited a markedly greater embryo size and growth rate at the maturation stage (S4), resulting in a higher seed quality than those of SP1 and CN1. This suggests that PTT1 under high temperatures perhaps produces ethylene [36] in its embryo cells. This results in high DNA contents, increased embryo mitotic cell division and numerous complete embryo cells and leads to a high seed quality, as indicated by GI, AA and SGR.

The high temperatures indicated by a high AGDD led to a shorter rice life cycle, including shortened days to anthesis and also to embryo and endosperm development. The higher AGDD caused a decreased size and growth rate of the embryo and endosperm, consequently resulting in a reduced yield and reduced yield components, such as the 1000-seed weight, the total seed weight/pot and the filled seeds/pot, in all rice varieties (Table 4). Previous reports suggested that high temperatures led to a shortened anthesis period, resulting in the rice having poor anther dehiscence, increased sterile spikelets (high temperatures at >35 °C, [16]; high night temperature of 32 °C, Mohammed and Tarpley [37]; [29]), decreased fertile spikelets (high night temperature at 24–35 °C, [38]), reduced pollination, less pollen tube development, less pollen germination [29] and reduced seed setting [10,16]. In addition, high temperatures reduced the grain size, amylose content and gel consistency [37], decreased the grain weight per panicle [38] and also reduced the grain yield [9]. The reduction in the rice yield components and yield was due to high temperatures.

In our results, high temperatures indicated by a high AGDD affected the grain and seed quality by increasing chalky grains and decreasing the GI, AA and SGR (Table 5). This result was similar to that of Cliu et al. [39], who reported that high temperatures stimulated increased chalky grains and a reduced grain length in rice. Chalky grain was an abnormality in the structural features of the starch granule of the rice grains, and it increased with increasing temperature during the milky stage development [40]. The starch composition during seed development was affected by high temperatures, resulting in a reduction in the amylose content and an alteration in the amylopectin structure as an indicator of abnormal starch synthesis and a key component of chalkiness in the rice grains [41].

The combination of nine traits from PCA analysis and eight traits from distinctive traits were considered as indicators of heat tolerance in three rice varieties (Figure 5 and Table 6). Mahendran et al. [42] reported that the traits from PCA analysis with high variability were expected to a provide high level of high temperature tolerance with high genetic variability in rice. Thus, we considered the anthesis day, AGDD of anthesis, total seed weight/panicle, 1000-seed weight, phenology (sowing-PM), filled seed/pot and SGR in determining the heat tolerance level. Consequently, the most heat tolerant variety was CN1, because it had the highest variation of those traits, followed by SP1. PTT1 was indicated to be heat sensitive.

## 4. Materials and Methods

### 4.1. Plant Materials

Three photoperiod-insensitive rice varieties (*Oryza sativa* L.), Pathum Thani 1 (PTT1), Suphan Buri 1 (SP1) and Chai Nat 1 (CN1), rice varieties often chosen for cultivation in the off-season and late off-season in Thailand, were used as plant materials in this study. The seeds were washed by soaking them in tap water for 10 min and then air dried. The dried seeds were mixed with 25% Metalaxyl (methyl-N-(methoxyacetyl)-N-(2,6-xylyl)-DL-alaninate) for sowing in this study.

### 4.2. Cultivating Conditions and High Temperature Treatment

The experiment was conducted at the Agronomy Experimental Station, Faculty of Agriculture, Khon Kaen University, Khon Kaen, Thailand (16°28′ N, 102°48′ E) in the off-season (PDI) and late off-season (PDII) from January to May in 2018 and 2019. The paddy soil properties in 2018 and 2019 were as follows: loam with N = 0.045%, P = 7.00 ppm, K = 49.00 ppm, pH = 3.91 and organic matter = 0.76% and loam with N = 0.035%, P = 6.00 ppm, K = 50.00 ppm, pH = 4.85 and organic matter = 0.71%, respectively. The experiment was designed in 2 × 3 factorials in a completely randomized design (CRD) with four replications. The factors for factorial design consisted of 2 planting dates, i.e., PDI (seed sowing on 15 January; a suitable time for rice grown in the off-season) and PDII (seed sowing on 1 February; late for rice grown in the off-season), and 3 rice varieties (PTT1, SP1 and CN1). The rice growing in both periods coincided with increased temperatures during summer in Thailand. Seeds mixed with 25% Metalaxyl were soaked with tap water for 24 h and then sown directly (with a seed rate of 125 kg/ha) in 0.8 m diameter cement pots containing 300 kg of paddy soil. The application of fertilizers consisted of N at the rate of 112.7 kg/ha, P at the rate of 18.75 kg/ha and K at the rate of 37.5 kg/ha, and the water level in the cement pots was maintained at 10 cm above the soil surface throughout the experiment.

### 4.3. Climate Data

Climate data such as maximum (T_max_), average (T_aver_) and minimum (T_min_) temperatures and relative humidity (RH) were recorded daily by the automatic weather station at the Agronomy Experimental Station, Khon Kaen University, Thailand in 2018 and 2019. The accumulated heat units, expressed as the accumulated growing degree day (AGDD), were calculated by the summation of the growing degree days (GDD) over the course of the rice growing time. GDD was calculated according to McMaster and Wilhelm [43] following Equation (1):GDD = (T_max_ + T_min_)/2 – T_base_(1)
where T_max_ and T_min_ are the daily maximum and minimum temperature, respectively, and T_base_ is the base temperature (T_base_ of rice = 10.0 °C).

### 4.4. Phenological Data

Rice phenological data was recorded throughout the experiment by visual daily observation such as germination, panicle initiation (PI), milky and physiological maturity (PM). The visual daily observation of the rice was evaluated based on the standard evaluation system for rice [44].

### 4.5. Embryo and Seed Development

At 0–15 days after anthesis, embryo development of ten-labeled seeds in each replication was determined using the plant microtechnique method [45,46]. The collected embryo samples were fixed in formaldehyde-alcohol-acetic acid (FAA) and then rinsed with 70% ethanol. The embryo samples were dehydrated by soaking, in order, in 50%, 70%, 85%, 95% and 100% ethanol for 120 min at each concentration. For the paraffin method, dehydrated embryo samples were infiltrated with melted paraffin at 57 °C for 24 h. The paraffinized rice embryos were embedded in a block with additional paraffin. The embedded rice embryos were sectioned to 15 μm using a rotary microtome (CUT4050 SLEE mainz, Micro Tec, Walldorf, Germany). The sections were deparaffinized by xylene for 24 h. The deparaffinized sections were stained in 1% Safranin followed by 1% Fast Green in 70% ethanol. The stained section was observed and photographed with a light microscope (CH2, Olympus, Tokyo, Japan). All images were analyzed using Adobe Photoshop software version PS.

At 0–25 days after anthesis, seed development (such as seed size) was measured from fifty labeled seeds in each replication that had their husks peeled. The size of the seeds was determined by photography with a stereo microscope (SZ, Olympus, Tokyo, Japan), and then all images were analyzed using Adobe Photoshop software version PS.

### 4.6. Yield and Yield Components

At physiological maturity (PM), yield components and yield parameters including number of panicles/pot, total seed weight/pot and percentage filled seed/pot were investigated for all plants in each pot. The number of panicles/plant, number of seeds/panicle, total seed weight/panicle and 1000-seed weight were collected from 20 plants in each pot.

### 4.7. Grain and Seed Quality

#### 4.7.1. Grain Quality

One hundred rice grains in each pot were harvested at PM and dried at room temperature for fourteen days or until reaching 14% moisture per grain for determining grain quality such as length, width, area and size of grain and also chalky grain. Length, width and areas of those grains were measured. Then, grain size was measured by photography with a stereo microscope (SZ, Olympus, Tokyo, Japan), and then all images were analyzed using Adobe Photoshop software version PS. Chalky grains were determined by photography with a stereo microscope (SZ, Olympus, Tokyo, Japan). Non-transparent areas of grain were interpreted as chalkiness, including grain parts of center, belly or back. A twenty-percent or greater non-transparent area of grain was determined as chalky grain [47].

#### 4.7.2. Seed Quality

Fifty rice seeds in each pot were harvested at PM and dried at room temperature for 14 days or until they reached 14% seed moisture for determining seed quality, such as germination percentage (G), germination index (GI), accelerated aging test (AA) and seedling growth rate (SGR).

##### Germination Percentage (G)

Seed germination test was conducted with 50 seeds per seed lot with 4 replications. It was measured using the Between paper method (BP) according to ISTA (2013). The rice seeds were placed in moistened rolled paper at 25 °C. Seeds were counted twice at 5 and 14 days and were considered germinated with normal seedlings. G was calculated as in Abeysiriwardena et al. [48] using the following Equation (2):(2)$$G\;(\%)=\left.\left(\;\frac{\mathrm{number}\;\mathrm{of}\;\mathrm{normal}\;\mathrm{seedlings}}{\mathrm{number}\;\mathrm{of}\;\mathrm{seed}\;\mathrm{tested}}\right.\right)\times 100$$

##### Germination Index (GI)

Germination index was calculated as described by Copeland and McDonald [49] following Equation (3):(3)$$\mathrm{GI}=\left.\left(\;\frac{\mathrm{number}\;\mathrm{of}\;\mathrm{germinated}\;\mathrm{seeds}}{\mathrm{days}\;\mathrm{to}\;\mathrm{first}\;\mathrm{count}}\right.\right)+\ldots +\left.\left(\;\frac{\mathrm{number}\;\mathrm{of}\;\mathrm{germinated}\;\mathrm{seeds}}{\mathrm{days}\;\mathrm{to}\;\mathrm{final}\;\mathrm{count}}\right.\right)$$

##### Accelerated Aging Test (AA)

An accelerated aging test was performed [50]. Fifty seeds were put on a wire mesh tray and put in a box with 40 mL tap water. Seeds were warmed and aged at 41 °C for 96 h. Aged seeds were sown on moistened rolled paper at 25 °C. The 14-days-normal seedlings were counted, and the number expressed as a percentage.

##### Seedling Growth Rate (SGR)

Fifty seeds were germinated, and the normal seedlings were counted. These normal seedlings were measured as the seedling growth rate (SGR). Then, they were placed in a paper bag and dried at 80.0 °C for 48 h. Dried normal seedlings were weighed to obtain their dry weight, and the SGR was calculated according to Copeland and McDonald [49] using following Equation (4):SGR = seedling dry weight/number of seedlings(4)

### 4.8. Statistical Analysis

The experiment was conducted using a completely randomized design (CRD). Data were subjected to three-way analysis of variance (ANOVA) to test the differences with the years, planting dates and varieties of yield, yield components, and grain and seed quality. Duncan’s multiple range test (DMRT) was used to compare yield, yield components, grain and seed quality in both years. A pairwise T-test comparison was performed for the years and planting dates and the phenology, embryo and seed development. A principle component analysis (PCA) was performed to reduce the dimensions of the multivariate data in PDII of 2019, including the phenology, embryo and seed development, yield, yield components, grain and seed quality. All statistical analyses were performed using SPSS software version 26.

## 5. Conclusions

In summary, high temperatures as indicated by high AGDD during the rice growing period in 2019 affected the rice phenology shift and resulted in early flowering. After anthesis, the high temperatures shortened the developmental period of the embryo and endosperm, leading to a smaller size and slower growth rate of the embryo and endosperm. High temperatures also caused a lower percentage of total seed weight/panicle, 1000-seed weight and filled seed/pot, resulting in yield reduction. In addition, high temperatures caused low seed quality, as indicated by a lower SGR. Moreover, high temperatures also led to a greater number of chalky grains. The phenology of rice var. PTT1 was sharply shortened, resulting in the smallest endosperm size, a low percentage of filled seed and a low seed weight. As a result, PTT1 was indicated to be a heat-sensitive rice variety. However, PTT1 exhibited slightly better seed germinability, probably due to its larger embryo size. In contrast, the phenology of CN1 was the least affected by the high AGDD, resulting in the highest percentage of filled seeds and seed weight. Thus, CN1 was indicated to be a heat tolerant rice variety, even though it showed a high percentage of chalky grain. We suggest that farmer select suitable rice varieties for the hot summer period.

## Figures and Tables

**Figure 1 plants-12-00666-f001:**
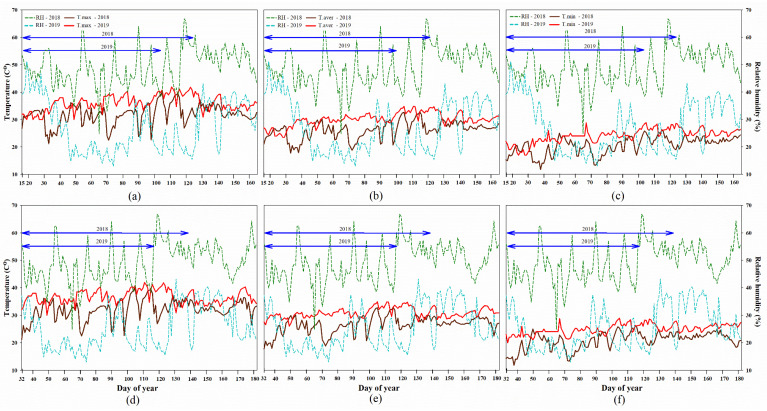
Climate data including daily maximum temperature (**a**,**d**), average temperature (**b**,**e**), minimum temperature (**c**,**f**) and relative humidity (**a**–**f**) during the period of rice growing (sowing-PM) on PDI (off-season, **a**–**c**) and PDII (late off-season, **d**–**f**) in 2018 and 2019. Blue lines indicate the rice growing period in 2018 (above) and 2019 (below).

**Figure 2 plants-12-00666-f002:**
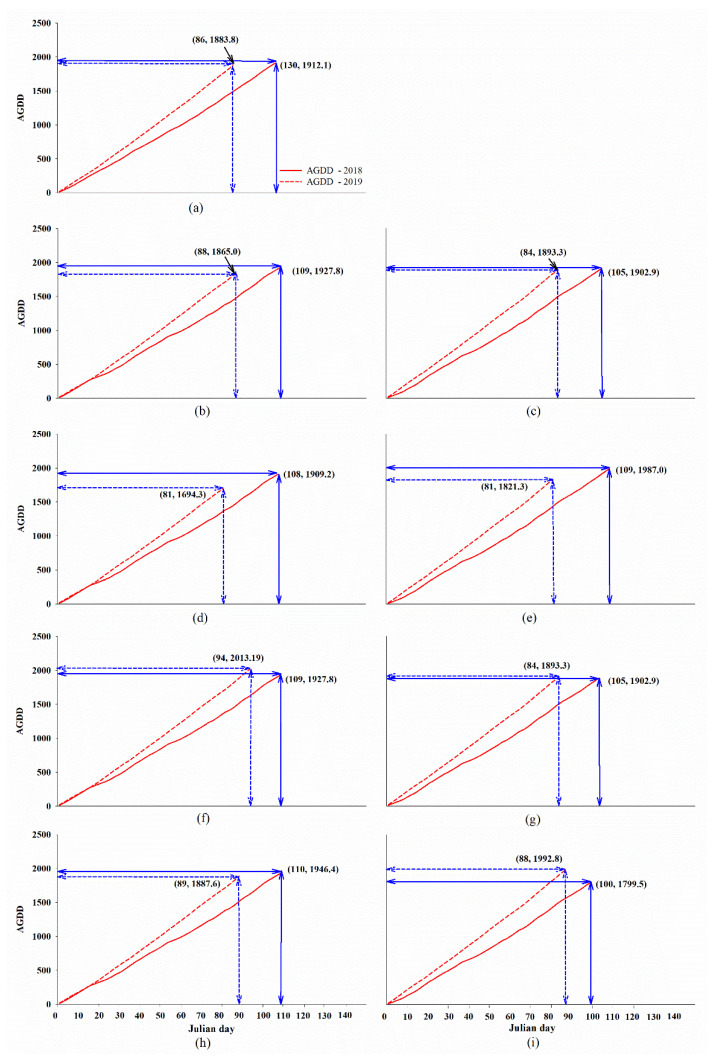
Accumulated growing degree days (AGDD) influenced rice growing period (sowing-PM). The total growing period of all rice affected by AGDD in 2018 and 2019 (**a**) and phenology shift in all rice cultivated on PDI (**b**) and PDII (**c**) in both years. Three rice varieties, Pathum Thani1 (**d**,**e**), Suphan Buri1 (**f**,**g**) and Chai Nat1 (**h**,**i**) were grown with different planting dates: PDI (off-season, d, f and h) and PDII (late off-season, (**e**,**g**,**i**) in 2018 and 2019. Red line and dashed red line indicate AGDD in 2018 and 2019, respectively. Blue line and dashed blue line indicate Julian day in 2018 and 2019, respectively.

**Figure 3 plants-12-00666-f003:**
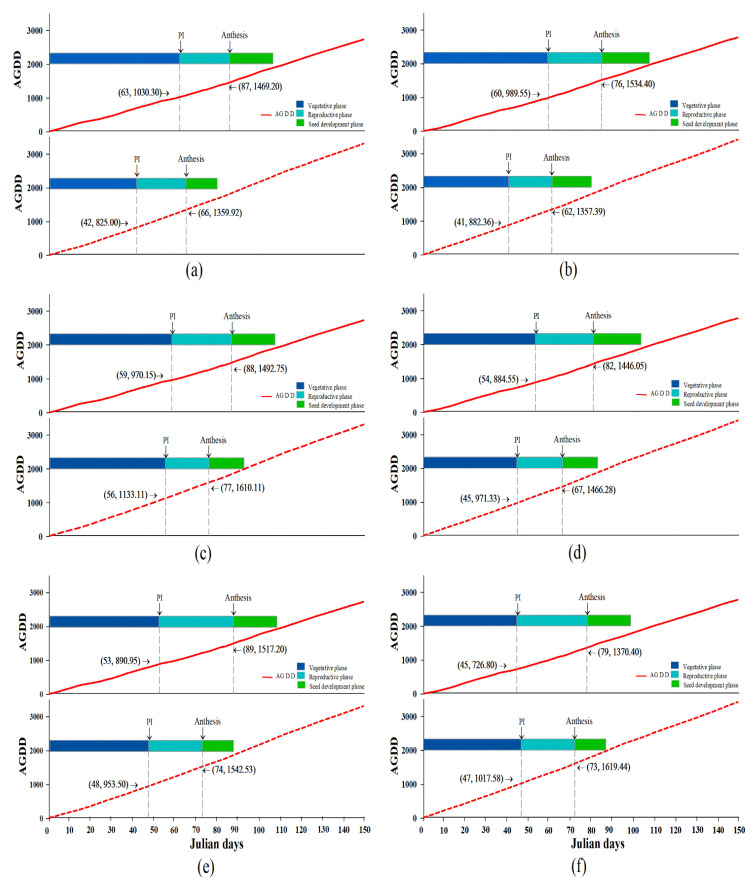
AGDD influenced changed growth stage (vegetative, reproductive and seed development phase) of rice vars. Pathum Thani1 (**a**,**b**), Suphan Buri 1 (**c**,**d**) and Chai Nat 1 (**e**,**f**) cultivated on PDI (off-season, (**a**,**c**,**e**)) and PDII (late off-season, (**b**,**d**,**f**) in 2018 (above in each Figure) and 2019 (below in each Figure). PI = panicle initiation. Red line and dashed red line indicate AGDD in PDI and PDII, respectively.

**Figure 4 plants-12-00666-f004:**
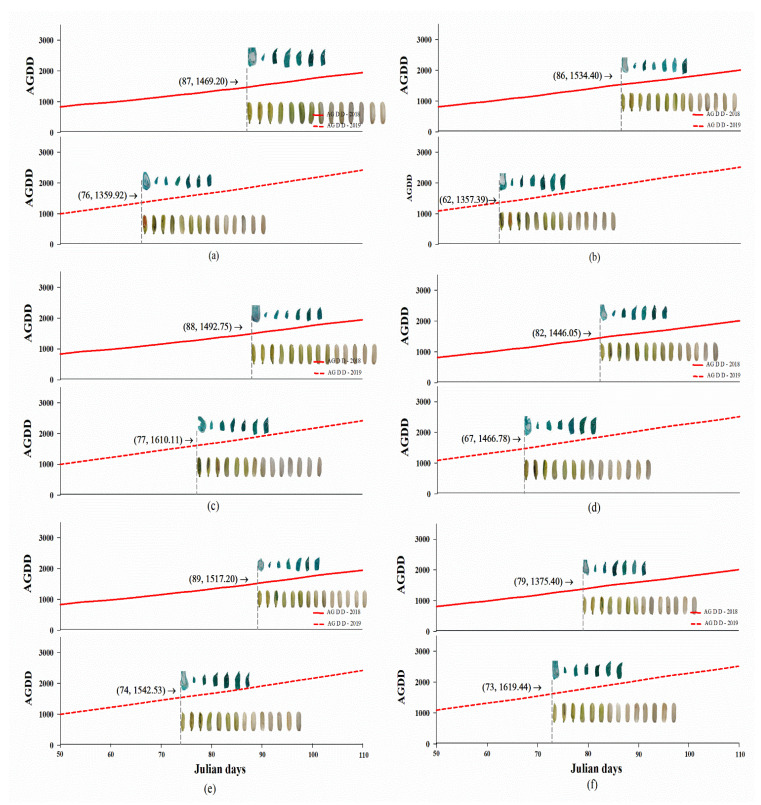
The increased AGDD influenced embryonic and seed development of rice vars. Pathum Thani1 (**a**,**b**), Suphan Buri1 (**c**,**d**) and Chai Nat1 (**e**,**f**) cultivated on PDI (off-season, (**a**,**c**,**e**)) and PDII (late off-season, (**b**,**d**,**f**)) in 2018 (above in each Figure) and in 2019 (below in each Figure). Red line and dashed red line indicate AGDD in PDI and PDII, respectively.

**Figure 5 plants-12-00666-f005:**
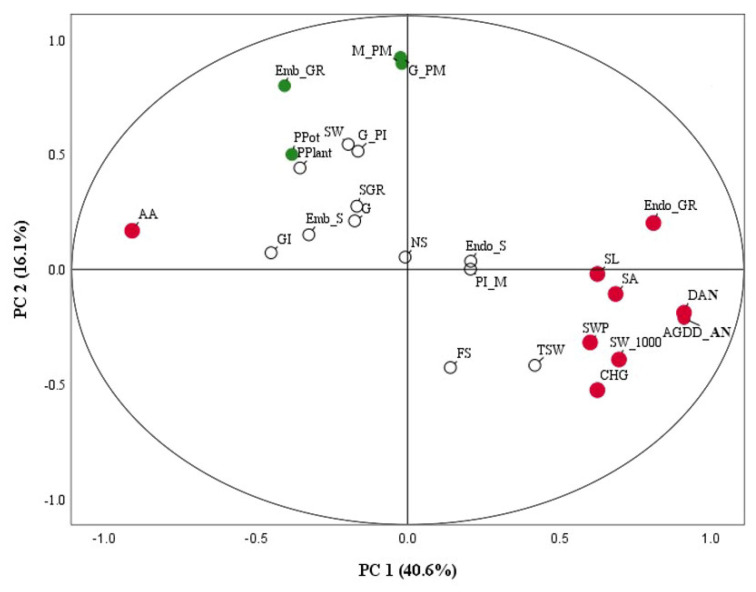
Principal component analysis (PCA) loading plot of phenology, grain and seed development, yield and yield components of three rice varieties under high temperatures. (Germination-PI = G-PI, PI-Milky = PI-M, Milky-PM = M-PM, Germination-PM = G-PM, Endosperm growth rate = Endo_GR, Embryo growth rate = Emb_GR, Endosperm size = Endo_S, Embryo size = Emb_S, Day of anthesis = DAN, AGDD of anthesis = AGDD_DAN, No. panicle/plant = PPlant, No. seed /panicle = NS, Total seed weight/panicle = SWP, 1000-seed weight = SW_1000, No. panicle/pot = PPot, Total seed weight/pot = TSW, Filled seed/pot = FS, Seed length = SL, Seed width = SW, Seed area = SA, Chalky grain = CHG, Germination percentage = G, Germination index = GI, Accelerating ageing = AA and Seedling growth rate = SGR). Red circles, green circles and white circles represent PC1, PC2 and PC3-6, respectively.

**Table 1 plants-12-00666-t001:** Phenology shift such as germination-panicle initiation (germination-PI), panicle initiation-milky (PI-milky) and milky-physiological maturity (milky-PM) of three rice varieties under different planting dates in 2018 and 2019 (means ± SE, *n* = 4).

Planting Date (PD)	Rice Varieties/Experimental Year											
	PTT1			SP1			CN1			All Varieties		
	2018	2019	*t*-Test	2018	2019	*t*-Test	2018	2019	*t*-Test	2018	2019	*t*-Test
	Germination-PI											
PDI	46.88 ± 0.63aA	43.75 ± 1.61aA	ns	43.13 ± 1.57aA	46.25 ± 0.72bA	ns	41.88 ± 0.63aA	41.25 ± 1.61aA	ns	43.96 ± 0.19aA	43.75 ± 0.19 aA	ns
PDII	45.63 ± 1.20aA	48.75 ± 1.61aA	ns	43.75 ± 0.72aB	51.88 ± 1.57aA	**	42.50 ± 1.02aA	45.63 ± 1.20aA	ns	43.96 ± 0.00 aA	48.75 ± 0.01 aA	ns
*t*-test	ns	ns		ns	*		ns	ns		ns	ns	
Average	46.25 ± 0.67A	46.25 ± 1.42A	ns	43.44 ± 0.81B	49.06 ± 1.33A	**	42.19 ± 0.57A	43.44 ± 1.24A	ns	43.96 ± 0.13A	46.25 ± 0.01A	ns
	PI-Milky											
PDI	38.25 ± 1.61aA	30.63 ± 1.20aB	**	38.88 ± 1.20aA	30.00 ± 1.02aB	**	37.00 ± 1.02aA	30.63 ± 1.57aB	*	38.04 ± 0.19 aA	30.42 ± 0.19 aA	ns
PDII	36.38 ± 0.63aA	27.50 ± 1.77aB	**	41.38 ± 1.20aA	28.13 ± 0.63aB	**	37.00 ± 1.02aA	30.63 ± 1.20aB	**	38.25 ± 0.00 aA	28.75 ± 0.01 aA	ns
*t*-test	ns	ns		ns	ns		ns	ns		ns	ns	
Average	37.31 ± 0.88A	29.06 ± 1.15B	**	40.13 ± 0.92A	29.06 ± 0.66B	**	37.00 ± 0.67A	30.63 ± 0.92B	**	38.15 ± 0.13A	29.58 ± 0.02A	ns
	Milky-PM											
PDI	15.25 ± 0.48aA	13.75 ± 0.48aA	ns	15.00 ± 0.41aA	12.25 ± 0.63aB	**	13.50 ± 0.65aA	10.00 ± 0.71aB	*	14.58 ± 0.19aA	12.00 ± 0.19aA	ns
PDII	14.00 ± 0.71aA	10.25 ± 0.48bB	**	13.50 ± 0.65aA	12.25 ± 0.25aA	ns	13.50 ± 0.87aA	10.00 ± 0.58aB	*	13.67 ± 0.01aA	10.83 ± 0.01aA	ns
*t*-test	ns	**		ns	ns		ns	ns		ns	ns	
Average	14.63 ± 0.46A	12.00 ± 0.73B	**	14.25 ± 0.45A	12.25 ± 0.31B	**	13.50 ± 0.50A	10.00 ± 0.42B	**	14.13 ± 0.13A	11.42 ± 0.01B	*
	Germination-PM											
PDI	100.38 ± 0.19aA	88.13 ± 0.19aB	*	97.00 ± 0.19bA	88.50 ± 0.19bB	*	92.38 ± 0.19bA	81.88 ± 0.19aA	ns	96.58 ± 0.19aA	86.17 ± 0.19aA	ns
PDII	96.00 ± 0.00bA	86.50 ± 0.00aA	ns	98.63 ± 0.05aA	92.25 ± 0.01aB	*	93.00 ± 0.00aA	86.25 ± 0.01aA	ns	95.88 ± 0.00bA	88.33 ± 0.00aA	ns
*t*-test	*	ns		*	*		*	ns		*	ns	
Average	98.19 ± 0.52A	87.31 ± 0.03B	**	97.81 ± 0.52A	90.38 ± 0.02B	**	92.69 ± 0.52A	84.06 ± 0.02B	**	96.23 ± 0.13A	87.25 ± 0.01B	**

Different capital letters within the same row show significant differences between planting dates by *t*-test (*p* ≤ 0.05). Different small letters within the same column show significant differences between years by *t*-test (*p* ≤ 0.05). ns: non-significant, * and **: significant difference at the level of *p* ≤ 0.05 and 0.01, respectively.

**Table 2 plants-12-00666-t002:** Anthesis day and AGDD after sowing related embryonic and seed development of rice vars. Pathum Thani1, Suphan Buri1 and Chai Nat1 cultivated on PDI (off-season) and PDII (late off-season) in 2018 and in 2019 (means ± SE, *n* = 4).

Planting Date (PD)	Rice Varieties											
	PTT1			SP1			CN1			All Varieties		
	2018	2019	*t*-Test	2018	2019	*t*-Test	2018	2019	*t*-Test	2018	2019	*t*-Test
	Day of anthesis beginning embryonic and seed development (DAS)											
PD1	87.25 ± 2.02aA	66.25 ± 2.06aB	**	87.75 ± 0.75aA	77.00 ± 0.00aB	**	89.00 ± 1.68aA	73.50 ± 1.50aB	**	88.00 ± 0.85aA	72.25 ± 1.55aB	**
PD2	85.75 ± 1.89aA	61.75 ± 0.75aB	**	82.25 ± 2.59aA	67.25 ± 0.85bB	**	79.25 ± 2.69bA	72.50 ± 1.66aA	ns	82.42 ± 1.49bA	67.17 ± 1.46bB	**
*t*-test	ns	ns		ns	**		*	ns		**	*	
Average	86.50 ± 1.31A	64.00 ± 1.32B	**	85.00 ± 1.63A	72.13 ± 1.88B	**	84.13 ± 2.36A	73.00 ± 1.05B	**	85.21 ± 1.02A	69.71 ± 1.17B	**
	AGDD at anthesis day influenced embryonic and seed development after anthesis (^o^C)											
PD1	1480.76 ± 43.20aA	1366.08 ± 48.05aA	ns	1487.61 ± 17.68aB	1610.11 ± 0.00aA	**	1519.61 ± 37.20aA	1532.86 ± 32.69aA	ns	1496.00 ± 18.71aA	1503.02 ± 35.35aA	ns
PD2	1522.19 ± 37.13aA	1351.99 ± 15.86aB	**	1443.78 ± 53.57aA	1473.16 ± 20.19bA	ns	1385.18 ± 58.10aB	1604.96 ± 42.08aA	*	1450.38 ± 31.30aA	1476.70 ± 34.51aA	ns
*t*-test	ns	ns		ns	**		ns	ns		ns	ns	
Average	1501.48 ± 27.51A	1359.03 ± 23.57B	**	1465.69 ± 27.40A	1541.64 ± 27.52A	ns	1452.39 ± 40.81B	1568.91 ± 28.18A	*	1473.19 ± 18.45A	1489.86 ± 24.32A	ns

Different capital letters within the same row show significant differences between planting dates by *t*-test (*p* ≤ 0.05). Different small letters within the same column show significant differences between years by *t*-test (*p* ≤ 0.05). ns: non-significant, * and **: significant difference at the level of *p* ≤ 0.05 and 0.01, respectively.

**Table 3 plants-12-00666-t003:** Embryo size, embryo growth rate, endosperm size and endosperm growth rate of three rice varieties under different planting dates (PDI and PDII) in 2018 and 2019 (means ± SE, *n* = 4).

Planting Date (PD)	Rice Varieties											
	PTT1			SP1			CN1			All Varieties		
	2018	2019	*t*-Test	2018	2019	*t*-Test	2018	2019	*t*-Test	2018	2019	*t*-Test
	embryo size (mm^2^/seed)											
PD1	3.55 ± 0.04aA	3.05 ± 0.05aB	**	3.32 ± 0.09aA	3.06 ± 0.06aB	*	3.23 ± 0.08aA	2.94 ± 0.05bB	*	3.37 ± 0.06aA	3.01 ± 0.01bA	ns
PD2	3.56 ± 0.14aA	3.15 ± 0.06aB	*	3.40 ± 0.10aA	3.12 ± 0.04aB	*	3.34 ± 0.09aA	3.15 ± 0.04aA	ns	3.43 ± 0.07aA	3.14 ± 0.02aA	ns
*t*-test	ns	ns		ns	ns		ns	*		ns	*	
Average	3.56 ± 0.07A	3.10 ± 0.04B	**	3.36 ± 0.07A	3.09 ± 0.03B	**	3.29 ± 0.06A	3.04 ± 0.05B	**	3.40 ± 0.05A	3.08 ± 0.03B	**
	embryo growth rate (mm^2^/day)											
PD1	0.29 ± 0.01aA	0.25 ± 0.01aB	**	0.27 ± 0.01aA	0.24 ± 0.01aB	*	0.27 ± 0.01aA	0.24 ± 0.01aB	*	0.28 ± 0.01aA	0.25 ± 0.01bB	**
PD2	0.29 ± 0.01aA	0.26 ± 0.01aB	*	0.28 ± 0.01aA	0.26 ± 0.01aA	ns	0.28 ± 0.01aA	0.24 ± 0.01aB	**	0.28 ± 0.01aA	0.25 ± 0.01aB	**
*t*-test	ns	ns		ns	ns		ns	ns		ns	*	
Average	0.29 ± 0.01A	0.26 ± 0.01B	**	0.28 ± 0.01A	0.25 ± 0.01B	**	0.27 ± 0.01A	0.24 ± 0.01B	**	0.28 ± 0.01A	0.25 ± 0.01B	**
	endosperm size (mm^2^/seed)											
PD1	16.05 ± 0.89aA	14.76 ± 0.43aA	ns	16.62 ± 0.30aA	13.16 ± 0.16aB	**	16.29 ± 0.42aA	13.17 ± 0.69aB	**	16.32 ± 0.34aA	13.70 ± 0.29aB	**
PD2	12.81 ± 0.26bA	13.19 ± 0.26bA	ns	13.36 ± 0.75bA	13.64 ± 0.19aA	ns	13.81 ± 0.24bA	14.23 ± 0.52aA	ns	13.33 ± 0.28bB	13.69 ± 0.21aA	*
*t*-test	*	*		**	ns		**	ns		**	ns	
Average	14.43 ± 0.75A	13.97 ± 0.38A	ns	14.99 ± 0.72A	13.40 ± 0.15B	*	15.05 ± 0.52A	13.70 ± 0.45A	ns	14.82 ± 0.60A	13.69 ± 0.17A	ns
	endosperm growth rate (mm^2^/day)											
PD1	0.60 ± 0.04aA	0.46 ± 0.01aB	**	0.64 ± 0.01aA	0.47 ± 0.03aB	**	0.63 ± 0.02aA	0.47 ± 0.01aB	**	0.62 ± 0.01aA	0.48 ± 0.02aB	**
PD2	0.47 ± 0.01bA	0.43 ± 0.01aB	*	0.51 ± 0.03bA	0.53 ± 0.02aA	ns	0.53 ± 0.01bA	0.51 ± 0.04aA	ns	0.50 ± 0.01bA	0.49 ± 0.01a	ns
*t*-test	*	ns		**	ns		**	ns		**	ns	
Average	0.53 ± 0.03A	0.44 ± 0.01B	*	0.57 ± 0.03A	0.52 ± 0.02A	ns	0.58 ± 0.02A	0.49 ± 0.02B	**	0.56 ± 0.02A	0.48 ± 0.01B	**

Different capital letters within the same row show significant differences between planting dates by *t*-test (*p* ≤ 0.05). Different small letters within the same column show significant differences between years by *t*-test (*p* ≤ 0.05). ns: non-significant, * and **: significant difference at the level of *p* ≤ 0.05 and 0.01, respectively.

**Table 4 plants-12-00666-t004:** Effect of year, planting dates and varieties on yield and yield components on PDI and PDII of rice off-season from January to May 2018 and 2019 (means ± SE, *n* = 4–24).

Factors	No. Panicle/Pot	No. Panicle/Plant	No. Seed /Panicle	Filled Seed/Pot (%)	Total Seed Weight/Pot (g)	Total Seed Weight/ Panicle	1000-Seed Weight (g)
Year (Y)							
2018	291.29 ± 4.39^b^	7.20 ± 0.17	54.88 ± 1.55^b^	96.32 ± 0.33^a^	295.93 ± 7.73^a^	1.26 ± 0.04^a^	26.65 ± 0.22^a^
2019	309.25 ± 10.75^a^	7.07 ± 0.24	61.17 ± 1.81^a^	67.00 ± 1.45^b^	172.25 ± 4.56^b^	0.59 ± 0.02^b^	24.22 ± 0.31^b^
F-test	*	ns	**	**	**	**	**
Planting date (PD)							
PDI	310.17 ± 10.29^a^	7.34 ± 0.18^a^	55.92 ± 1.34^b^	81.43 ± 3.06	234.52 ± 13.70	0.90 ± 0.08	25.88 ± 0.40^a^
PDII	290.38 ± 5.24^b^	6.93 ± 0.23^b^	60.13 ± 2.09^a^	81.90 ± 3.39	233.66 ± 15.01	0.93 ± 0.07	24.99 ± 0.31^b^
F-test	*	*	*	ns	ns	ns	**
Variety (V)							
PTT1	315.06 ± 11.43^a^	7.67 ± 0.22^a^	59.31 ± 2.01	81.22 ± 4.12^b^	248.82 ± 18.80^a^	0.93 ± 0.11	24.64 ± 0.37^b^
SP1	315.25 ± 9.46^a^	7.63 ± 0.13^a^	59.81 ± 2.15	78.91 ± 4.42^b^	223.95 ± 18.39^b^	0.90 ± 0.10	24.93 ± 0.51^b^
CN1	270.50 ± 4.15^b^	6.11 ± 0.18^b^	54.94 ± 2.34	84.85 ± 3.22^a^	229.50 ± 15.35^ab^	0.95 ± 0.08	26.74 ± 0.24^a^
F-test	**	**	ns	**	*	ns	**
Y × PD							
2018 × PDI	286.75 ± 4.47^b^	7.49 ± 0.20	57.83 ± 2.10^b^	95.37 ± 0.39	293.27 ± 10.69	1.28 ± 0.05^a^	27.29 ± 0.32^a^
2018 × PDII	295.83 ± 7.55^b^	6.90 ± 0.27	51.92 ± 2.03^b^	97.28 ± 0.39	298.58 ± 11.57	1.24 ± 0.06^a^	26.01 ± 0.18^b^
2019 × PDI	333.58 ± 17.97^a^	7.19 ± 0.31	54.00 ± 1.56^b^	67.49 ± 1.93	175.76 ± 6.56	0.54 ± 0.02^b^	24.46 ± 0.46^b^
2019 × PDII	284.92 ± 7.24^b^	6.95 ± 0.38	68.33 ± 1.39^a^	66.51 ± 2.23	168.75 ± 6.47	0.64 ± 0.04^b^	23.97 ± 0.42^c^
F-test	**	ns	**	ns	ns	*	*
Y × V							
2018 × PTT1	290.38 ± 6.59^a^	7.55 ± 0.36^ab^	57.88 ± 3.16	96.66 ± 0.37^a^	316.87 ± 11.30	1.33 ± 0.07^a^	26.01 ± 0.08^b^
2018 × SP1	305.00 ± 7.57^a^	7.49 ± 0.18^ab^	56.00 ± 2.36	95.32 ± 0.75^a^	288.42 ± 13.14	1.24 ± 0.06^a^	26.63 ± 0.50^b^
2018 × CN1	278.50 ± 6.27^b^	6.55 ± 0.21^b^	50.75 ± 2.09	96.99 ± 0.43^a^	282.49 ± 13.84	1.22 ± 0.07^a^	27.32 ± 0.33^a^
2019 × PTT1	339.75 ± 18.52^a^	7.79 ± 0.27^a^	60.75 ± 2.59	65.79 ± 2.09^bc^	180.76 ± 8.00	0.54 ± 0.03^c^	23.27 ± 0.21^c^
2019 × SP1	325.50 ± 17.21^a^	7.76 ± 0.19^a^	63.63 ± 3.17	62.49 ± 2.50^c^	159.48 ± 9.43	0.55 ± 0.05^c^	23.22 ± 0.19^c^
2019 × CN1	262.50 ± 4.03^c^	5.66 ± 0.19^c^	59.13 ± 3.75	72.72 ± 1.49^b^	176.52 ± 4.01	0.68 ± 0.03^b^	26.16 ± 0.19^b^
F-test	**	*	ns	*	ns	*	**
PD × V							
PDI × PTT1	337.25 ± 19.39	8.03 ± 0.18	58.50 ± 2.16	79.73 ± 6.18	261.44 ± 26.53	0.94 ± 0.17	24.70 ± 0.46^d^
PDI × SP1	321.88 ± 17.65	7.66 ± 0.14	58.13 ± 1.93	78.76 ± 5.94	218.82 ± 22.25	0.86 ± 0.15	25.62 ± 0.87^c^
PDI × CN1	271.38 ± 4.04	6.34 ± 0.21	51.13 ± 2.08	85.79 ± 3.84	223.29 ± 22.25	0.92 ± 0.12	27.32 ± 0.32^a^
PDII × PTT1	292.88 ± 6.62	7.31 ± 0.37	60.13 ± 3.52	82.71 ± 5.81	236.19 ± 27.67	0.92 ± 0.15	24.57 ± 0.61^d^
PDII × SP1	308.63 ± 7.72	7.59 ± 0.23	61.50 ± 3.91	79.06 ± 6.96	229.08 ± 30.77	0.93 ± 0.13	24.23 ± 0.46^d^
PDII × CN1	269.63 ± 7.57	5.88 ± 0.28	58.75 ± 3.87	83.92 ± 5.43	235.72 ± 22.44	0.97 ± 0.11	26.17 ± 0.21^b^
F-test	ns	ns	ns	ns	ns	ns	**
Y × PD × V							
2018 × PDI × PTT1	291.00 ± 5.21	7.95 ± 0.33	62.00 ± 3.19	95.78 ± 0.26	327.73 ± 16.62	1.37 ± 0.09	25.84 ± 0.10^bc^
2018 × PDI × SP1	294.50 ± 9.13	7.70 ± 0.25	59.75 ± 2.98	94.40 ± 1.04	273.35 ± 11.64	1.25 ± 0.08	27.90 ± 0.13^a^
2018 × PDI × CN1	274.75 ± 5.94	6.83 ± 0.17	51.75 ± 3.15	95.93 ± 0.22	278.75 ± 15.73	1.22 ± 0.11	28.14 ± 0.13^a^
2018 × PDII × PTT1	289.75 ± 13.23	7.15 ± 0.63	53.75 ± 5.01	97.55 ± 0.18	306.01 ± 15.52	1.28 ± 0.13	26.17 ± 0.06^b^
2018 × PDII × SP1	315.50 ± 10.51	7.28 ± 0.25	52.25 ± 2.78	96.25 ± 1.00	303.50 ± 22.77	1.22 ± 0.11	25.35 ± 0.27^c^
2018 × PDII × CN1	282.25 ± 11.78	6.28 ± 0.37	49.75 ± 3.15	98.04 ± 0.24	286.22 ± 25.23	1.22 ± 0.10	26.50 ± 0.23^b^
2019 × PDI × PTT1	383.50 ± 17.35	8.10 ± 0.21	55.00 ± 1.87	63.69 ± 2.57	195.16 ± 8.83	0.51 ± 0.04	23.57 ± 0.36^d^
2019 × PDI × SP1	349.25 ± 29.50	7.63 ± 0.15	56.50 ± 2.60	63.12 ± 0.88	164.29 ± 13.88	0.47 ± 0.03	23.33 ± 0.25^d^
2019 × PDI × CN1	268.00 ± 5.76	5.85 ± 0.17	50.50 ± 3.18	75.65 ± 0.63	167.83 ± 3.61	0.63 ± 0.01	26.49 ± 0.15^b^
2019 × PDII × PTT1	296.00 ± 4.80	7.48 ± 0.48	66.50 ± 2.40	67.88 ± 3.29	166.37 ± 9.11	0.56 ± 0.04	22.97 ± 0.14^d^
2019 × PDII × SP1	301.75 ± 11.66	7.90 ± 0.35	70.75 ± 2.53	61.87 ± 5.31	154.67 ± 14.38	0.63 ± 0.08	23.12 ± 0.31^d^
2019 × PDII × CN1	257.00 ± 4.74	5.48 ± 0.34	67.75 ± 2.43	69.79 ± 2.07	185.21 ± 3.42	0.72 ± 0.04	25.84 ± 0.27^bc^
F-test	ns	ns	ns	ns	ns	ns	**
CV (%)	2.22	7.67	3.78	4.62	5.62	6.36	2.20

Different small letters in the same column indicate significant difference between treatments by DMRT (*p* ≤ 0.05). ns: non-significant, * and **: significant difference at the level of *p* ≤ 0.05 and 0.01, respectively.

**Table 5 plants-12-00666-t005:** Effect of planting dates and varieties on grain and seed quality in rice off-season 2018 and 2019 (means ± SE, *n* = 4–24).

Factors	Length (mm.)	Width (mm.)	Seed Area (mm^2^)	Chalky Grain (%)	G (%)	GI	AA (%)	SGR (%)
Year (Y)								
2018	2.38 ± 0.07	7.43 ± 0.06^a^	13.79 ± 0.12^a^	28.75 ± 2.32^b^	93.83 ± 1.06^a^	12.28 ± 0.19^a^	95.67 ± 0.91^a^	9.50 ± 0.23^a^
2019	2.37 ± 0.02	7.30 ± 0.03^b^	13.38 ± 0.09^b^	60.92 ± 6.25^a^	85.33 ± 2.63^b^	10.82 ± 0.38^b^	79.00 ± 2.88^b^	7.92 ± 0.10^b^
F-test	ns	*	**	**	**	**	**	**
Planting date (PD)								
PDI	2.35 ± 0.02	7.32 ± 0.03	13.51 ± 0.10	54.08 ± 6.17^a^	94.42 ± 0.86^a^	12.45 ± 0.15^a^	93.08 ± 1.30^a^	8.97 ± 0.22^a^
PDII	2.41 ± 0.07	7.40 ± 0.06	13.66 ± 0.12	35.58 ± 4.63^b^	84.75 ± 2.61^b^	10.65 ± 0.37^b^	81.58 ± 3.26^b^	8.45 ± 0.25^b^
F-test	ns	ns	ns	**	**	**	**	**
Variety (V)								
PTT1	2.42 ± 0.10	7.48 ± 0.07^a^	13.48 ± 0.17^b^	39.44 ± 5.47	93.25 ± 2.17	12.28 ± 0.32^a^	92.25 ± 2.47^a^	9.07 ± 0.35^a^
SP1	2.32 ± 0.02	7.40 ± 0.04^a^	13.42 ± 0.12^b^	43.13 ± 5.29	89.75 ± 2.89	11.10 ± 0.40^b^	88.88 ± 2.93^a^	8.64 ± 0.21^b^
CN1	2.39 ± 0.02	7.20 ± 0.03^b^	13.86 ± 0.09^a^	51.94 ± 9.52	85.75 ± 2.67	11.26 ± 0.46^b^	80.88 ± 3.96^b^	8.42 ± 0.29^b^
F-test	ns	**	**	ns	ns	**	**	**
Y × PD								
2018 × PDI	2.32 ± 0.02	7.35 ± 0.04	13.61 ± 0.13	26.50 ± 2.80^b^	95.17 ± 1.42^a^	12.78 ± 0.24^a^	96.83 ± 0.63^a^	9.87 ± 0.17^a^
2018 × PDII	2.44 ± 0.14	7.50 ± 0.10	13.98 ± 0.18	31.00 ± 3.71^b^	92.50 ± 1.54^a^	11.78 ± 0.23^a^	94.50 ± 1.67^a^	9.13 ± 0.40^b^
2019 × PDI	2.38 ± 0.04	7.29 ± 0.04	13.41 ± 0.15	81.67 ± 3.61^a^	93.67 ± 0.98^a^	12.12 ± 0.14^a^	89.33 ± 2.04^a^	8.06 ± 0.17^c^
2019 × PDII	2.37 ± 0.01	7.31 ± 0.05	13.35 ± 0.12	40.17 ± 8.50^b^	77.00 ± 3.90^b^	9.52 ± 0.53^b^	68.67 ± 3.35^b^	7.78 ± 0.11^c^
F-test	ns	ns	ns	**	**	**	**	*
Y x V								
2018 × PTT1	2.51 ± 0.21	7.58 ± 0.14	13.79 ± 0.31	32.75 ± 2.72^b^	98.50 ± 0.63	13.03 ± 0.34	98.75 ± 0.53^a^	10.38 ± 0.12^a^
2018 × SP1	2.30 ± 0.01	7.45 ± 0.06	13.71 ± 0.13	37.50 ± 2.13^b^	92.50 ± 1.80	11.57 ± 0.29	95.75 ± 0.80^ab^	9.44 ± 0.10^b^
2018 × CN1	2.34 ± 0.02	7.25 ± 0.04	13.88 ± 0.14	16.00 ± 2.14^c^	90.50 ± 1.59	12.23 ± 0.12	92.50 ± 2.10^abc^	8.68 ± 0.53^c^
2019 × PTT1	2.33 ± 0.02	7.38 ± 0.02	13.17 ± 0.07	46.13 ± 10.39^b^	88.00 ± 3.46	11.54 ± 0.40	85.75 ± 3.71^bc^	7.76 ± 0.15^d^
2019 × SP1	2.34 ± 0.04	7.36 ± 0.05	13.12 ± 0.13	48.75 ± 10.31^b^	87.00 ± 5.50	10.64 ± 0.74	82.00 ± 4.77^c^	7.84 ± 0.06^d^
2019 × CN1	2.45 ± 0.02	7.16 ± 0.04	13.84 ± 0.12	87.88 ± 3.88^a^	81.00 ± 4.64	10.29 ± 0.78	69.25 ± 4.91^d^	8.16 ± 0.25^cd^
F-test	ns	ns	ns	**	ns	ns	**	**
PD × V								
PDI × PTT1	2.31 ± 0.03	7.36 ± 0.03	13.17 ± 0.09^b^	50.88 ± 8.74	96.50 ± 1.45	12.82 ± 0.41	97.00 ± 1.07	9.02 ± 0.51^ab^
PDI × SP1	2.32 ± 0.04	7.38 ± 0.06	13.37 ± 0.16^ab^	54.75 ± 7.64	95.75 ± 0.88	12.13 ± 0.14	94.00 ± 1.41	8.49 ± 0.30^b^
PDI × CN1	2.42 ± 0.03	7.23 ± 0.04	13.99 ± 0.11^a^	56.63 ± 15.43	91.00 ± 1.36	12.40 ± 0.11	88.25 ± 2.81	9.38 ± 0.28^a^
PDII × PTT1	2.54 ± 0.20	7.61 ± 0.13	13.79 ± 0.31^ab^	28.00 ± 3.80	90.00 ± 3.89	11.74 ± 0.43	87.50 ± 4.31	9.11 ± 0.52^ab^
PDII × SP1	2.32 ± 0.02	7.42 ± 0.06	13.46 ± 0.18^ab^	31.50 ± 4.78	83.75 ± 4.96	10.08 ± 0.61	83.75 ± 5.23	8.79 ± 0.31^b^
PDII × CN1	2.36 ± 0.02	7.18 ± 0.04	13.73 ± 0.13^ab^	47.25 ± 12.01	80.50 ± 4.56	10.13 ± 0.72	73.50 ± 6.61	7.46 ± 0.14^c^
F-test	ns	ns	**	ns	ns	ns	ns	**
Y × PD × V								
2018 × PDI × PTT1	2.30 ± 0.04	7.35 ± 0.05	13.18 ± 0.11	28.50 ± 3.30^bc^	99.00 ± 1.00	13.50 ± 0.61	99.50 ± 0.50	10.33 ± 0.20^a^
2018 × PDI × SP1	2.32 ± 0.02	7.41 ± 0.09	13.65 ± 0.18	35.00 ± 2.89^b^	96.50 ± 1.50	12.31 ± 0.10	95.50 ± 0.50	9.26 ± 0.15^cd^
2018 × PDI × CN1	2.36 ± 0.04	7.29 ± 0.05	14.00 ± 0.20	16.00 ± 2.16^d^	90.00 ± 2.16	12.52 ± 0.08	95.50 ± 0.50	10.03 ± 0.24^ab^
2018 × PDII × PTT1	2.72 ± 0.41	7.81 ± 0.22	14.40 ± 0.44	37.00 ± 3.42^b^	98.00 ± 0.82	12.56 ± 0.17	98.00 ± 0.82	10.43 ± 0.17^a^
2018 × PDII × SP1	2.28 ± 0.01	7.48 ± 0.07	13.78 ± 0.19	40.00 ± 2.94^b^	88.50 ± 1.50	10.83 ± 0.13	96.00 ± 1.63	9.61 ± 0.06^bc^
2018 × PDII × CN1	2.32 ± 0.02	7.20 ± 0.07	13.75 ± 0.19	16.00 ± 4.08^d^	91.00 ± 2.65	11.94 ± 0.04	89.50 ± 3.77	7.34 ± 0.19^f^
2019 × PDI × PTT1	2.31 ± 0.03	7.36 ± 0.04	13.16 ± 0.16	73.25 ± 3.47^a^	94.00 ± 2.16	12.15 ± 0.32	94.50 ± 0.96	7.72 ± 0.23^ef^
2019 × PDI × SP1	2.33 ± 0.08	7.35 ± 0.08	3.09 ± 0.19	74.50 ± 2.02^a^	95.00 ± 1.00	11.95 ± 0.24	92.50 ± 2.75	7.72 ± 0.06^ef^
2019 × PDI × CN1	2.49 ± 0.02	7.16 ± 0.05	13.98 ± 0.12	97.25 ± 2.43^a^	92.00 ± 1.83	12.27 ± 0.19	81.00 ± 1.29	8.74 ± 0.21^d^
2019 × PDII × PTT1	2.35 ± 0.01	7.41 ± 0.02	13.18 ± 0.03	19.00 ± 1.29^cd^	82.00 ± 5.23	10.93 ± 0.62	77.00 ± 3.51	7.80 ± 0.23^ef^
2019 × PDII × SP1	2.35 ± 0.02	7.37 ± 0.08	13.15 ± 0.22	23.00 ± 7.05^cd^	79.00 ± 9.88	9.33 ± 1.17	71.50 ± 4.99	7.97 ± 0.05^e^
2019 × PDII × CN1	2.41 ± 0.02	7.16 ± 0.06	13.71 ± 0.20	78.50 ± 2.40^a^	70.00 ± 4.08	8.31 ± 0.46	57.50 ± 4.35	7.59 ± 0.20^ef^
F-test	ns	ns	ns	**	ns	ns	ns	**
CV (%)	9.55	1.50	1.53	20.62	3.06	6.77	3.92	6.18

Different small letters in the same column indicate significant difference between treatments by DMRT (*p* ≤ 0.05). ns: non-significance, * and **: significant difference at the level of *p* ≤ 0.05 and 0.01, respectively. G: Germination percentage, GI: Germination index, AA: Accelerated aging, SGR: Seedling growth rate.

**Table 6 plants-12-00666-t006:** High temperature tolerant traits were performed by PCA and distinctive traits of three rice varieties. (means ± SE, *n* = 4–24).

High Temperature Tolerant Traits Considered by PCA (no. 1–9) and Distinctive Traits (no. 10–17)	PTT1	SP1	CN1
1. Anthesis day	64.00 ± 1.32^b^	72.13 ± 1.88^a^	73.00 ± 1.05^a^
2. AGDD of anthesis	1359.03 ± 23.57^c^	1541.64 ± 27.52^b^	1568.91 ± 28.18^a^
3. Endosperm growth rate	0.44 ± 0.01^c^	0.51 ± 0.02^a^	0.49 ± 0.02^b^
4. Total seed weight/panicle	0.54 ± 0.03^b^	0.55 ± 0.05^b^	0.68 ± 0.03^a^
5. 1000-seed weight	23.27 ± 0.21^b^	23.22 ± 0.19^b^	26.16 ± 0.19^a^
6. Seed length,	ns	ns	ns
7. Seed area	ns	ns	ns
8. Chalky grain	46.13 ± 10.39^b^	48.75 ± 10.31^b^	87.88 ± 3.88^a^
9. Accelerating ageing	85.75 ± 3.71^a^	82.00 ± 4.77^a^	69.25 ± 4.91^b^
10. Phenology (Sowing-PM)	81.00 ± 1.04^c^	84.13 ± 0.45^b^	87.63 ± 0.60^a^
11. Embryo size	ns	ns	ns
12. Embryo growth rate	ns	ns	ns
13. Endosperm size	ns	ns	ns
14. No. panicle/pot	339.75 ± 18.52^a^	325.50 ± 17.21^a^	262.50 ± 4.03^b^
15. No. panicle/plant	7.79 ± 0.27^a^	7.76 ± 0.19^a^	5.66 ± 0.19^b^
16. Filled seed/pot	65.79 ± 2.09^b^	62.49 ± 2.50^c^	72.72 ± 1.49^a^
17. SGR	7.76 ± 0.15^b^	7.84 ± 0.06^b^	8.16 ± 0.25^a^

Different small letters in the same column indicate significant difference between treatments by DMRT (*p* ≤ 0.05). ns: non-significance at the level of *p* ≤ 0.05. Seedling growth rate = SGR and physiological maturity = PM.

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
