# Peer review of "High Temperature Alters Phenology, Seed Development and Yield in Three Rice Varieties"

_plants, 2023, doi:10.3390/plants12030666_

Round 1

Reviewer 1 Report

Dear Dr. Editor

The manuscript has a certain amount of work and some practical innovation, but the text is not well arranged. Except English writing and presentation, some mistakes were found. Please find my comments in the following.

1)       Title “High temperature in late off-season alters phenology, seed development, and yield in rice”

“Phenology”: The scientific study of periodic biological phenomena, such as flowering, breeding, and migration, in relation to climatic conditions. Phenology is not proper for this paper, please change or delete it

And suggest “high temperature” change to “high temperature of natural field”

2)       In this paper, “high temperature” is indicted by higher accumulated growing degree day during the whole rice growing period.

In generally, high temperature data are collected from initiation stage and continued to maturity covering the whole reproductive stage. Please change the whole temperature conditions data to the whole reproductive stage data.

3)”Abstract”

“Among varieties, high temperature in 2019 caused the smallest phenological shift in Chai Nat 1(CN1) while
the shift was largest in Pathum Thani 1 (PTT1). In addition, CN1 exhibited significantly higher 1000 seed weight and percent filled seeds. It was suggested that farmers avoid growing rice during PDII due to the risk of high temperature – induced reduction in grain yield and quality

It was suggested that farmers avoid growing rice during PDII due to the risk of high temperature

This conclusion made in this abstract is not supported by the data logically. What was for 2018 in PDII?

4) “Results

2.1 climate data”

Please adjust “climate data “to “4. Materials and methods”, and describe these climate data simply.

  5) “Results and discussion

   These parts don’t clearly, succinctly and reasonably analysis and interpretation of the data

In Table3 and 4, please add the SD of the concerned parameters,

And comparisons were made between concerned characteristics in the plant PDII and PDI?

Author Response

22nd January 2023

Dear Reviewer 1

Thank you so much for valuable suggestions of our MS and we are pleased to submit the revised version of plants-2161525 entitled “High temperature in late off-season alters phenology, seed development and yield in rice” which changed to “High temperature alters phenology, seed development and yield in three rice-varieties” as the present form for intended publication in Plants for your kind consideration. We appreciated the constructive criticism of the reviewers. As comments of reviewers, we tried to revised carefully following reviewer’s suggestions in track changes for adding more information in the text. Our responses were given point by point as outlined below.

Sincerely,

Anoma Dongsansuk      

The Responses to Editor and reviewers ’s comments

Reviewer 1’s comments

1) Title “High temperature in late off-season alters phenology, seed development, and yield in rice”

“Phenology”: The scientific study of periodic biological phenomena, such as flowering, breeding, and migration, in relation to climatic conditions. Phenology is not proper for this paper, please change or delete it

And suggest “high temperature” change to “high temperature of natural field”

Thank you very much for your valuable suggestions. For phenology, we found high temperature indicated by AGDD, stimulated early flower formation and early flowering in all rice varieties. Thus, we would use phenology in title. For high temperature in title, we would say “high temperature” in title because we did our experiment in the pot under natural field, we did not do experiment in the exactly field. And we also deleted “in late off-season” in title according results.  

2) In this paper, “high temperature” is indicted by higher accumulated growing degree day during the whole rice growing period. In generally, high temperature data are collected from initiation stage and continued to maturity covering the whole reproductive stage. Please change the whole temperature conditions data to the whole reproductive stage data.

Thank you very much for your valuable suggestions. Actually, we collected temperature data for AGDD from seed sowing to physiological maturity (PM).

3) ”Abstract”

“Among varieties, high temperature in 2019 caused the smallest phenological shift in Chai Nat 1(CN1) while
the shift was largest in Pathum Thani 1 (PTT1). In addition, CN1 exhibited significantly higher 1000 seed weight and percent filled seeds. It was suggested that farmers avoid growing rice during PDII due to the risk of high temperature – induced reduction in grain yield and quality” “It was suggested that farmers avoid growing rice during PDII due to the risk of high temperature”. This conclusion made in this abstract is not supported by the data logically. What was for 2018 in PDII?

From the revised results by three way ANOVA, we found planting dates influenced phenology, size and growth rate of endosperm, and seed quality in different rice varieties but they did not influence yield. So we prefer 2019 indicated as higher temperature condition than in 2018 not including both planting date (PDI and PDII). And we revised the text in whole MS including abstract according revised results.

4) “Results

2.1 climate data”

Please adjust “climate data “to “4. Materials and methods”, and describe these climate data simply.

Thank you very much for your valuable suggestion. We would present climate data in result part because it‘s important to support temperature data in different planting dates and years according growing period, seed and yield parameters.  

5) “Results and discussion

These parts don’t clearly, succinctly and reasonably analysis and interpretation of the data

We tried to analyze and generate a new result such as in Table 2 and 6 and Figure 7 and analyzed data in three way ANOVA as shown in Table 4 and 5, and then we rewrote results and discussions for the whole MS by simplified and proofreading by native speaker as attached proofreading certificate file.     

In Table3 and 4, please add the SD of the concerned parameters,

Thank you very much for your valuable suggestions. The results of Table 1 and 2, we added SE values so we would add SE values in Table 3 and 4 for unity of the whole MS.

And comparisons were made between concerned characteristics in the plant PDII and PDI?

We already made compared between PDII and PDI by t-test as shown in Table 1-3.

22nd January 2023

Dear Reviewer 1

Thank you so much for valuable suggestions of our MS and we are pleased to submit the revised version of plants-2161525 entitled “High temperature in late off-season alters phenology, seed development and yield in rice” which changed to “High temperature alters phenology, seed development and yield in three rice-varieties” as the present form for intended publication in Plants for your kind consideration. We appreciated the constructive criticism of the reviewers. As comments of reviewers, we tried to revised carefully following reviewer’s suggestions in track changes for adding more information in the text. Our responses were given point by point as outlined below.

Sincerely,

   Anoma Dongsansuk      

The Responses to Editor and reviewers ’s comments

Reviewer 1’s comments

1) Title “High temperature in late off-season alters phenology, seed development, and yield in rice”

“Phenology”: The scientific study of periodic biological phenomena, such as flowering, breeding, and migration, in relation to climatic conditions. Phenology is not proper for this paper, please change or delete it

And suggest “high temperature” change to “high temperature of natural field”

Thank you very much for your valuable suggestions. For phenology, we found high temperature indicated by AGDD, stimulated early flower formation and early flowering in all rice varieties. Thus, we would use phenology in title. For high temperature in title, we would say “high temperature” in title because we did our experiment in the pot under natural field, we did not do experiment in the exactly field. And we also deleted “in late off-season” in title according results.  

2) In this paper, “high temperature” is indicted by higher accumulated growing degree day during the whole rice growing period. In generally, high temperature data are collected from initiation stage and continued to maturity covering the whole reproductive stage. Please change the whole temperature conditions data to the whole reproductive stage data.

Thank you very much for your valuable suggestions. Actually, we collected temperature data for AGDD from seed sowing to physiological maturity (PM).

3) ”Abstract”

“Among varieties, high temperature in 2019 caused the smallest phenological shift in Chai Nat 1(CN1) while
the shift was largest in Pathum Thani 1 (PTT1). In addition, CN1 exhibited significantly higher 1000 seed weight and percent filled seeds. It was suggested that farmers avoid growing rice during PDII due to the risk of high temperature – induced reduction in grain yield and quality” “It was suggested that farmers avoid growing rice during PDII due to the risk of high temperature”. This conclusion made in this abstract is not supported by the data logically. What was for 2018 in PDII?

From the revised results by three way ANOVA, we found planting dates influenced phenology, size and growth rate of endosperm, and seed quality in different rice varieties but they did not influence yield. So we prefer 2019 indicated as higher temperature condition than in 2018 not including both planting date (PDI and PDII). And we revised the text in whole MS including abstract according revised results.

4) “Results

2.1 climate data”

Please adjust “climate data “to “4. Materials and methods”, and describe these climate data simply.

Thank you very much for your valuable suggestion. We would present climate data in result part because it‘s important to support temperature data in different planting dates and years according growing period, seed and yield parameters.  

5) “Results and discussion

These parts don’t clearly, succinctly and reasonably analysis and interpretation of the data

We tried to analyze and generate a new result such as in Table 2 and 6 and Figure 7 and analyzed data in three way ANOVA as shown in Table 4 and 5, and then we rewrote results and discussions for the whole MS by simplified and proofreading by native speaker as attached proofreading certificate file.     

In Table3 and 4, please add the SD of the concerned parameters,

Thank you very much for your valuable suggestions. The results of Table 1 and 2, we added SE values so we would add SE values in Table 3 and 4 for unity of the whole MS.

And comparisons were made between concerned characteristics in the plant PDII and PDI?

We already made compared between PDII and PDI by t-test as shown in Table 1-3.

22nd January 2023

Dear Reviewer 1

Thank you so much for valuable suggestions of our MS and we are pleased to submit the revised version of plants-2161525 entitled “High temperature in late off-season alters phenology, seed development and yield in rice” which changed to “High temperature alters phenology, seed development and yield in three rice-varieties” as the present form for intended publication in Plants for your kind consideration. We appreciated the constructive criticism of the reviewers. As comments of reviewers, we tried to revised carefully following reviewer’s suggestions in track changes for adding more information in the text. Our responses were given point by point as outlined below.

Sincerely,

   Anoma Dongsansuk      

The Responses to Editor and reviewers ’s comments

Reviewer 1’s comments

1) Title “High temperature in late off-season alters phenology, seed development, and yield in rice”

“Phenology”: The scientific study of periodic biological phenomena, such as flowering, breeding, and migration, in relation to climatic conditions. Phenology is not proper for this paper, please change or delete it

And suggest “high temperature” change to “high temperature of natural field”

Thank you very much for your valuable suggestions. For phenology, we found high temperature indicated by AGDD, stimulated early flower formation and early flowering in all rice varieties. Thus, we would use phenology in title. For high temperature in title, we would say “high temperature” in title because we did our experiment in the pot under natural field, we did not do experiment in the exactly field. And we also deleted “in late off-season” in title according results.  

2) In this paper, “high temperature” is indicted by higher accumulated growing degree day during the whole rice growing period. In generally, high temperature data are collected from initiation stage and continued to maturity covering the whole reproductive stage. Please change the whole temperature conditions data to the whole reproductive stage data.

Thank you very much for your valuable suggestions. Actually, we collected temperature data for AGDD from seed sowing to physiological maturity (PM).

3) ”Abstract”

“Among varieties, high temperature in 2019 caused the smallest phenological shift in Chai Nat 1(CN1) while
the shift was largest in Pathum Thani 1 (PTT1). In addition, CN1 exhibited significantly higher 1000 seed weight and percent filled seeds. It was suggested that farmers avoid growing rice during PDII due to the risk of high temperature – induced reduction in grain yield and quality” “It was suggested that farmers avoid growing rice during PDII due to the risk of high temperature”. This conclusion made in this abstract is not supported by the data logically. What was for 2018 in PDII?

From the revised results by three way ANOVA, we found planting dates influenced phenology, size and growth rate of endosperm, and seed quality in different rice varieties but they did not influence yield. So we prefer 2019 indicated as higher temperature condition than in 2018 not including both planting date (PDI and PDII). And we revised the text in whole MS including abstract according revised results.

4) “Results

2.1 climate data”

Please adjust “climate data “to “4. Materials and methods”, and describe these climate data simply.

Thank you very much for your valuable suggestion. We would present climate data in result part because it‘s important to support temperature data in different planting dates and years according growing period, seed and yield parameters.  

5) “Results and discussion

These parts don’t clearly, succinctly and reasonably analysis and interpretation of the data

We tried to analyze and generate a new result such as in Table 2 and 6 and Figure 7 and analyzed data in three way ANOVA as shown in Table 4 and 5, and then we rewrote results and discussions for the whole MS by simplified and proofreading by native speaker as attached proofreading certificate file.     

In Table3 and 4, please add the SD of the concerned parameters,

Thank you very much for your valuable suggestions. The results of Table 1 and 2, we added SE values so we would add SE values in Table 3 and 4 for unity of the whole MS.

And comparisons were made between concerned characteristics in the plant PDII and PDI?

We already made compared between PDII and PDI by t-test as shown in Table 1-3.

22nd January 2023

Dear Reviewer 1

Thank you so much for valuable suggestions of our MS and we are pleased to submit the revised version of plants-2161525 entitled “High temperature in late off-season alters phenology, seed development and yield in rice” which changed to “High temperature alters phenology, seed development and yield in three rice-varieties” as the present form for intended publication in Plants for your kind consideration. We appreciated the constructive criticism of the reviewers. As comments of reviewers, we tried to revised carefully following reviewer’s suggestions in track changes for adding more information in the text. Our responses were given point by point as outlined below.

Sincerely,

   Anoma Dongsansuk      

The Responses to Editor and reviewers ’s comments

Reviewer 1’s comments

1) Title “High temperature in late off-season alters phenology, seed development, and yield in rice”

“Phenology”: The scientific study of periodic biological phenomena, such as flowering, breeding, and migration, in relation to climatic conditions. Phenology is not proper for this paper, please change or delete it

And suggest “high temperature” change to “high temperature of natural field”

Thank you very much for your valuable suggestions. For phenology, we found high temperature indicated by AGDD, stimulated early flower formation and early flowering in all rice varieties. Thus, we would use phenology in title. For high temperature in title, we would say “high temperature” in title because we did our experiment in the pot under natural field, we did not do experiment in the exactly field. And we also deleted “in late off-season” in title according results.  

2) In this paper, “high temperature” is indicted by higher accumulated growing degree day during the whole rice growing period. In generally, high temperature data are collected from initiation stage and continued to maturity covering the whole reproductive stage. Please change the whole temperature conditions data to the whole reproductive stage data.

Thank you very much for your valuable suggestions. Actually, we collected temperature data for AGDD from seed sowing to physiological maturity (PM).

3) ”Abstract”

“Among varieties, high temperature in 2019 caused the smallest phenological shift in Chai Nat 1(CN1) while
the shift was largest in Pathum Thani 1 (PTT1). In addition, CN1 exhibited significantly higher 1000 seed weight and percent filled seeds. It was suggested that farmers avoid growing rice during PDII due to the risk of high temperature – induced reduction in grain yield and quality” “It was suggested that farmers avoid growing rice during PDII due to the risk of high temperature”. This conclusion made in this abstract is not supported by the data logically. What was for 2018 in PDII?

From the revised results by three way ANOVA, we found planting dates influenced phenology, size and growth rate of endosperm, and seed quality in different rice varieties but they did not influence yield. So we prefer 2019 indicated as higher temperature condition than in 2018 not including both planting date (PDI and PDII). And we revised the text in whole MS including abstract according revised results.

4) “Results

2.1 climate data”

Please adjust “climate data “to “4. Materials and methods”, and describe these climate data simply.

Thank you very much for your valuable suggestion. We would present climate data in result part because it‘s important to support temperature data in different planting dates and years according growing period, seed and yield parameters.  

5) “Results and discussion

These parts don’t clearly, succinctly and reasonably analysis and interpretation of the data

We tried to analyze and generate a new result such as in Table 2 and 6 and Figure 7 and analyzed data in three way ANOVA as shown in Table 4 and 5, and then we rewrote results and discussions for the whole MS by simplified and proofreading by native speaker as attached proofreading certificate file.     

In Table3 and 4, please add the SD of the concerned parameters,

Thank you very much for your valuable suggestions. The results of Table 1 and 2, we added SE values so we would add SE values in Table 3 and 4 for unity of the whole MS.

And comparisons were made between concerned characteristics in the plant PDII and PDI?

We already made compared between PDII and PDI by t-test as shown in Table 1-3.

Reviewer 2 Report

General Comments:

Interesting manuscript about how temperature via AGDD affects development and yield in multiple rice varieties. The manuscript is at times confusing, with too much detailed description while containing little interpretation and synthesis of the findings and broader applications.

Firstly, the introduction is missing justifications about the aims, primarily due to the ‘on-season’ and off-season dichotomy not being adequately detailed. If rice is usually grown in the ‘on-season’ then why is the study interested in growing rice in the off-season? It might provide additional clarity if the authors explained the reasoning behind growing rice in the off-season, and how much of annual production or similar that comes from these conditions. Additionally, the actual ‘on-season’ is not properly explained, when is this in time?

Secondly, it is not clear from the way the discussion section is written what is original thought, interpretation of the findings or inference from other studies. Additionally, the discussion focuses almost completely on how temperature influences physiological processes, which is strictly not measured or observed by the authors, only the results and final steps of the processes. Therefore, the authors need to reformat the discussion by comparing the study findings to other similar data, while excluding for example how temperature influence photosynthesis and ATP, which has not been measured.

Finally, continuing with the discussion, the interpretation of how differences between years (2018 v 2019), planting dates (PDI v PDII) influence the different varieties (PTT1, SP1 and CN1) is mostly overlooked. What is the most important: yearly variation or variation within years? And how do the different varieties manage to adapt or produce within this variation? Is one variety better at handling one type of variation over the other, and are there trade-offs between different traits measured? These types of broader applications of the findings are completely forgotten while only prioritizing the effect of temperature on physiological processes.

Specific Comments:

1.     Lines 76-115. This entire section needs a re-write. I am not sure what the issue is, but I think the authors have confused DOY with Julian days (days since sowing or similar). This is apparent since the Tmax for PDI and PDII is the same but differs in dates somehow. In other places it differs but the higher temperature occurs earlier for PDI than PDII, which is not possible, both ‘treatments’ should experience the higher earlier temperature.

2.     Lines 231-245. This section also needs a re-write, the text does not make a lot of sense and no coherent message can be extracted from the results. Perhaps it is the sentence structure and wording that is the issue.

Figures and Tables:

1.     Figure 1. The X-axis of the figure makes the graph difficult to interpret. It would be a lot easier for the reader if the X-axis is DOY rather than days since sowing, or similar. Also, the blue lines are not explained properly.

2.     Figure 3. Missing legend information about the differentiation of years.

3.     Figure 4. It is not clear what this figure shows. The figure text specifies embryonic and seed development, but the text suggests date of anthesis. Either way, the information does not warrant a figure but can easily be put in a table if relevant.

4.     Figures 5 and 6 (and inset in Figure 4). The information from these figures are not used in the manuscript, therefore it would be better to put them in supplementary, or put the actual numbers gained from these observations in a table.

Author Response

22nd January 2023

Dear Reviewer 2

Thank you so much for valuable suggestions of our MS and we are pleased to submit the revised version of plants-2161525 entitled “High temperature in late off-season alters phenology, seed development and yield in rice” which changed to “High temperature alters phenology, seed development and yield in three rice-varieties” as the present form for intended publication in Plants for your kind consideration. We appreciated the constructive criticism of the reviewers. As comments of reviewers, we tried to revised carefully following reviewer’s suggestions in track changes for adding more information in the text. Our responses were given point by point as outlined below.

Sincerely,

Anoma Dongsansuk      

The Responses to Editor and reviewers ’s comments

Reviewer 2’s comments

General Comments:

  1. Interesting manuscript about how temperature via AGDD affects development and yield in multiple rice varieties. The manuscript is at times confusing, with too much detailed description while containing little interpretation and synthesis of the findings and broader applications.

We revised the whole MS by carefully and we deleted some sentences that looked more detailed and added more interpretation the results.  

  1. Firstly, the introduction is missing justifications about the aims, primarily due to the ‘on-season’ and off-season dichotomy not being adequately detailed. If rice is usually grown in the ‘on-season’ then why is the study interested in growing rice in the off-season? It might provide additional clarity if the authors explained the reasoning behind growing rice in the off-season,

Because “rice growing in Thailand, there are 2 cropping periods, namely the on-season and the off-season. The cultivation of off-season rice becomes rising popular in Thailand that increases by 42.9% in off-season rice area during in 2016-2020. In 2020, rice yield in on-season in Thailand was 26,423,822 tons and in the off-season was 4,553,778 tons. Particularly, in the Northeastern of Thailand, the rice growing area in off-season increased by 65.4%. Because the cultivation rice in off-season has high potential to increase rice yield, with rice yields approximately 631 kg/rai by 43.4% which is higher than on-season rice (approximately 440 kg/rai).

So we added these reasons in the introduction part in line 44-53.

and how much of annual production or similar that comes from these conditions. Additionally, the actual ‘on-season’ is not properly explained, when is this in time?

“For the annual production of off-season rice in Thailand is approximately 4,553,778 tons that is higher than in-season rice (26,423,822 tons). And rice growing in off-season is during January – April of year in Thailand.”

We added this sentence in introduction part in line 44-49.  

  1. Secondly, it is not clear from the way the discussion section is written what is original thought, interpretation of the findings or inference from other studies.

We rewrote the discussion part by interpreted from result and comparison our results with other studies.

Additionally, the discussion focuses almost completely on how temperature influences physiological processes, which is strictly not measured or observed by the authors, only the results and final steps of the processes. Therefore, the authors need to reformat the discussion by comparing the study findings to other similar data, while excluding for example how temperature influence photosynthesis and ATP, which has not been measured.

Done

  1. Finally, continuing with the discussion, the interpretation of how differences between years (2018 v 2019), planting dates (PDI v PDII) influence the different varieties (PTT1, SP1 and CN1) is mostly overlooked. What is the most important: yearly variation or variation within years?

From our results, we found yearly influenced the different varieties by changed phenology, size and growth rate of embryo and endosperm, grain and seed quality and yield. And year 2019 (indicated by AGDD) was indicated as higher temperature than year 2018. We found planting dates influenced phenology, size and growth rate of endosperm, and seed quality in different rice varieties but they did not influence yield. Thus, we indicated yearly variation is the most important.

And how do the different varieties manage to adapt or produce within this variation?

We found CN1 is the most heat tolerance then following by SP1 and PTT1 is heat sensitive. Therefore, it should select suitable rice varieties for hot summer period such as CN1 and/or SP1.

Is one variety better at handling one type of variation over the other,

Yes, it is CN1 that is better than other and it is indicated as heat tolerance by less affected phenology, the highest percentage of filled seeds and seed weight.

and are there trade-offs between different traits measured? These types of broader applications of the findings are completely forgotten while only prioritizing the effect of temperature on physiological processes.

Yes, there are some parameters such as anthesis day, AGDD of anthesis, total seed weight/panicle, 1000-seed weight, phenology (sowing-PM), filled seed/pot, SGR and percentage of chalky grain.

Specific Comments:

  1. Lines 76-115. This entire section needs a re-write. I am not sure what the issue is, but I think the authors have confused DOY with Julian days (days since sowing or similar). This is apparent since the Tmax for PDI and PDII is the same but differs in dates somehow. In other places it differs but the higher temperature occurs earlier for PDI than PDII, which is not possible, both ‘treatments’ should experience the higher earlier temperature.

We rewrote the text by simplified. And we changed Julian day to Day of year (DOY) in Figure 1 and we deleted the date in text to avoid confusing. Therefore, we presented only high temperature in PDI and PDII in each year.

  1. Lines 231-245. This section also needs a re-write, the text does not make a lot of sense and no coherent message can be extracted from the results. Perhaps it is the sentence structure and wording that is the issue.

We rewrote the text by simplified and proofreading by native speaker as attached proofreading certificate file.   

Figures and Tables:

  1. Figure 1. The X-axis of the figure makes the graph difficult to interpret. It would be a lot easier for the reader if the X-axis is DOY rather than days since sowing, or similar. Also, the blue lines are not explained properly.

We changed x-axis from Julian day to Day of year (DOY) and added the explanation of blue lines in legend of Figure 1.

  1. Figure 3. Missing legend information about the differentiation of years.

We added the legend of different of year as “in 2018 (above in each Figure) and 2019 (below in each Figure)”.

  1. Figure 4. It is not clear what this figure shows. The figure text specifies embryonic and seed development, but the text suggests date of anthesis. Either way, the information does not warrant a figure but can easily be put in a table if relevant.

We added the result of anthesis day and AGDD at anthesis day related embryonic and seed development of rice in Table 2.

  1. Figures 5 and 6 (and inset in Figure 4). The information from these figures are not used in the manuscript, therefore it would be better to put them in supplementary, or put the actual numbers gained from these observations in a table.

We moved Figure 5 and 6 to supplementary.

Reviewer 3 Report

plants-2161525/Article

Title:

“High temperature in late off-season alters phenology, seed development, and yield in rice”

Dear editor/author

The present manuscript presents useful research data. Climate change is very important for the agriculture. Temperature affects grain quality and sterility, photosynthesis rate and seed germination. Hence rice production can be reduced.

Introduction is clear and has background information.

Material and methods are well organized.

All the data well explained and discussed.

Discussion is too extensive.

References are recent.

However, there are some suggestions:

The most important:

1.      Regarding the title: I think title should be more specific: “High temperature in late off-season alters phenology, seed development, and yield in three rice-varieties”

2.      Keywords: 2-3 more keywords more should be involved

3.      Table 1,2: I don’t; t think that ‘all varieties’ should be included. The manuscript contains three distinctive varieties. Why the combined average should be included? (The same question for Figure 2)

4.      There are a lot of data. I strongly suggest using a PCA analysis; it could assist in visualizing the data.

5.      I am wondering why you did not perform a three-way analysis: Year (2018, 2019) × Planting date (PDI, PDII) × Rice variety (PTT1, SP1, CN1). I think that a three-way analysis could present more clear the data (non-so extensive).  

6.      Conclusions: I think you could involve your suggestions for the farmers. This section is disproportionate to the length of the discussion. I think also you should also try to reduce the discussion.

Revision list (text format)

1.      l. 33: write 10.0%-25.5%. Decimal numbers should be used everywhere in the text.

2.      l. 83: ‘+31.6 °C’. I don; think that ‘+’ needs; otherwise use it everywhere in the text.

3.      l. 83: insert ‘°’ from symbols. The correct is 40.0 °C (not 40.0oC) (also use space after ‘40.0’) .

4.      l. 114: “….but RH was lower ….”. I think ‘but’ should be replaced by ‘and’.

5.      l. 128: Use capital letters in the figures

6.      l. 150: The paragraph space should be checked

7.      l. 16219: The paper is extensive. Hence, I suggest merge some figures from Figure 2 and Figure 3.

8.      l. 229: Correct italics

9.      l. 247: Align the text

10.   l. 266: Use please ‘P’ or ‘p’ everywhere (check for italics all the tables)

11.   l. 288: correct the reference

12.   l. 386-387. No ‘space’ before ‘%’

13.   l. 416: move ‘FAA’ after parenthesis

14.   l. 419: click a ‘space’ after 57

15.   l. 453: Correct paragraph number

Author Response

22nd January 2023

Dear Reviewer 3

Thank you so much for valuable suggestions of our MS and we are pleased to submit the revised version of plants-2161525 entitled “High temperature in late off-season alters phenology, seed development and yield in rice” which changed to “High temperature alters phenology, seed development and yield in three rice-varieties” as the present form for intended publication in Plants for your kind consideration. We appreciated the constructive criticism of the reviewers. As comments of reviewers, we tried to revised carefully following reviewer’s suggestions in track changes for adding more information in the text. Our responses were given point by point as outlined below.

Sincerely,

Anoma Dongsansuk      

The Responses to Editor and reviewers ’s comments

Reviewer 3’s comments

1.Regarding the title: I think title should be more specific: “High temperature in late off-season alters phenology, seed development, and yield in three rice-varieties”

We added “three rice-varieties” in title.

2.Keywords: 2-3 more keywords more should be involved

We added 2 keywords such as embryonic development and seed quality.

3.Table 1,2: I don’t; t think that ‘all varieties’ should be included. The manuscript contains three distinctive varieties. Why the combined average should be included? (The same question for Figure 2)

Thank you very much for valuable suggestions. We would include all varieties results in Table 1-2 and Figure 2 because all varieties data can show the result trend of all rice and easier tell the trend of result between different variations.

4.There are a lot of data. I strongly suggest using a PCA analysis; it could assist in visualizing the data.

We did PCA analysis as your suggestion as shown in Figure 7.

5.I am wondering why you did not perform a three-way analysis: Year (2018, 2019) × Planting date (PDI, PDII) × Rice variety (PTT1, SP1, CN1). I think that a three-way analysis could present more clear the data (non-so extensive).  

We did follow as your suggestion by three-way ANOVA analysis with Year × PD × V in Table 4 and 5.

6.Conclusions: I think you could involve your suggestions for the farmers. This section is disproportionate to the length of the discussion.

We added suggestion for farmer in conclusions.

I think also you should also try to reduce the discussion.

Done

Revision list (text format)

  1. 33: write 10.0%-25.5%. Decimal numbers should be used everywhere in the text.

Done

  1. 83: ‘+31.6 °C’. I don; think that ‘+’ needs; otherwise use it everywhere in the text.

Done

  1. 83: insert ‘°’ from symbols. The correct is 40.0 °C (not 40.0oC) (also use space after ‘40.0’).

Done

  1. 114: “but RH was lower ….”. I think ‘but’ should be replaced by ‘and’.

Done

  1. 128: Use capital letters in the figures

From Plants format, it suggests to use small letters. Thus, we changed small letter in every Figures and also legend of Figures.

  1. 150: The paragraph space should be checked

Done

  1. 16219: The paper is extensive. Hence, I suggest merge some figures from Figure 2 and Figure 3.

Thank you very much for your valuable suggestion. For Figure 2, we would compare AGDD between year 2018 and 2019 influenced phenology (whole growing period from sowing to PM) in each planting date and it should be easier to compare AGDD between year 2018 and 2019 in the same Figure. But for Figure 3, we separate Figure in each year and planting date because we would present AGDD influenced growing period in sowing-PI, PI-anthesis and anthesis-PM and it is easier to see the phenology shift in each growth stage when compares between year 2018 (above Figure) and 2019 (below Figure). Thus, we would separate Figure 2 and 3.       

  1. 229: Correct italics

Done

  1. 247: Align the text

Done

  1. 266: Use please ‘P’ or ‘p’ everywhere (check for italics all the tables)

Done

1l.   l. 288: correct the reference

        Done

  1. 386-387. No ‘space’ before ‘%’

Done

  1. 416: move ‘FAA’ after parenthesis

Done

  1. 419: click a ‘space’ after 57

Done

  1. 453: Correct paragraph number

We deleted paragraph number because they are sub-heading of 4.7.2 of seed quality. So they don’t need to address paragraph number. 

Round 2

Reviewer 2 Report

General Comments:

The authors have responded adequately to the reviewer comments and the manuscript quality has improved with the additional textual edits and analyses.